# AutoScale: **Scale-Aware Data Mixing for Pre-Training LLMs**

**Feiyang Kang**[*][†]
Virginia Tech
fyk@vt.edu

**Yifan Sun**[*]
UIUC
yifan50@illinois.edu

**Bingbing Wen**
University of Washington
bingbw@uw.edu

**Si Chen**
Virginia Tech
chensi@vt.edu

**Dawn Song**
UC Berkeley
dawnsong@gmail.com

**Rafid Mahmood**
University of Ottawa & NVIDIA
mahmood@telfer.uottawa.ca

**Ruoxi Jia**[†]
Virginia Tech
ruoxijia@vt.edu

## Abstract

Domain reweighting is an emerging research area aimed at adjusting the relative weights of different data sources to improve the effectiveness and efficiency of LLM pre-training. We show that data mixtures that perform well at smaller scales may not retain their advantage at larger scales, challenging the existing practice of determining competitive mixtures in small-scale experiments and *directly* applying them at much larger scales. To address this, we propose AutoScale, a two-stage, scale-aware data composition framework. First, AutoScale fits a parametric model that predicts the model's loss under different data compositions, then uses it to find an approximate best allocation at smaller, more manageable budgets. Next, leveraging a novel theoretical analysis of how optimal compositions evolve with scale, AutoScale extrapolates that composition to larger budgets without further retraining. Empirically, AutoScale accelerates convergence and improves downstream performance. For instance, when pre-training GPT-2 Large, it achieves a 28% faster perplexity reduction than baselines and up to a 38% speed-up over unweighted training, while yielding best-average results on various downstream tasks. Overall, our findings illustrate how domain importance shifts with training scale, underscoring the need for scale-dependent data curation in LLM training. Our code is open-sourced[1].

## 1 Introduction

Large language models (LLMs) are pre-trained on vast datasets sourced from diverse domains. However, the immense computational demands of this process, coupled with limited resources, create a pressing need to enhance the effectiveness and efficiency of pre-training. A promising approach to address this challenge is through *domain reweighting*—adjusting the ratio (or weights) of data from different sources.

However, developing a principled and efficient framework for determining an optimal data mix remains challenging. Many industry pipelines still rely on trial-and-error heuristics (Rae et al., 2021; Grattafiori et al., 2024) or reuse domain weights designed for previous models (Mehta et al., 2024), without a systematic approach for deciding how much of each domain to include. The seminal domain-optimization work by Xie et al. (2024) attempted to upweight "difficult" domains, but later work (Fan et al., 2023) reported instability and only limited validation-loss improvements, partly because the chosen optimization objective does not robustly align with the model's ultimate test-time performance. Therefore, recent methods (Liu et al., 2024; Ye et al., 2024) focus on *directly* optimizing domain weights for lower validation loss. However, the highly complex relationship between domain weights and model performance makes such optimization expensive. A common strategy of these works is to reduce costs is to train multiple times at smaller scales, identify a "best" mix,

---

[*]Equal contribution. [†]Correspondence to: Feiyang Kang and Ruoxi Jia <fyk, ruoxijia>@vt.edu.
[1]https://github.com/feiyang-k/AutoScale

and then assume it transfers to large-scale pre-training . Yet our experiments show that compositions found at smaller scales may not remain competitive when training is scaled up, whereas directly optimizing at full scale is infeasible. This yields a *dilemma*: either accept small-scale solutions that may not transfer or attempt large-scale optimization that is prohibitively costly.

To resolve this dilemma, we develop a novel theoretical analysis that shows how the optimal domain composition evolves at different scales. Building on this insight, we propose a two-stage framework, AutoScale, for scale-aware domain reweighting. In the first stage, we fit a parametric model that predicts the model's loss under different data compositions, then use it to discover an approximate optimal allocation at smaller, more manageable budgets. Next, we apply our theoretical result to extrapolate that allocation to larger budgets—without re-optimizing at full scale—thus bridging the gap between small-scale optimization and full-scale pre-training.

Our experiments, conducted on both decoder-only and encoder-only architectures, consistently that AutoScale speeds up convergence and yields favorable downstream task performance. For instance, in pre-training GPT-2 Large on the RedPajama dataset, our approach achieves a 28% faster perplexity reduction compared with any baseline and up to a 38% speed-up over unweighted training, while also delivering the best downstream-task performance. Moreover, we made surprising empirical observations that data sources traditionally viewed as "high-quality" (e.g., Wikipedia and scientific papers) excel at smaller scales but exhibit sharp diminishing returns as the training grows. Meanwhile, domains containing more diverse examples (e.g., CommonCrawl) continue offering loss reductions at larger scales, underscoring the importance of scale-aware data curation.

## 2 Related Work

Principled training data curation for LLMs is an emerging research area, aiming to strategically select data that improves model performance. It can be performed at multiple levels—from token-level (Lin et al., 2024) or point-level (Wang et al., 2024) up to domain-level selection. Domain-level approaches are often more efficient because they operate at a coarser granularity, typically applying soft selection—i.e., upweighting or downweighting entire data domains.

Domain reweighting can generally be viewed as a two-step process: (i) define an objective that captures the goal of improving test loss or other model performance measures, and (ii) optimize domain weights according to that objective. DoReMi (Xie et al., 2024), a seminal paper in the space, adopted GroupDRO (Sagawa et al., 2019) as the objective, which implicitly upweights "difficult" domains. However, subsequent studies (Fan et al., 2023) found that the performance gains are unstable and limited, partly because the chosen objective does not align with the metrics ultimately used to evaluate the model at test time. Recent methods (Liu et al., 2024; Ye et al., 2024; Fan et al., 2023) attempt to directly optimize validation loss, which serves as a closer proxy for the test-time model performance metrics we actually care about. We note that there can still be misalignments between validation loss and test-time performance metrics (Barton, 2024), which is an active research area in itself, but validation loss remains a widely accepted objective for model selection in pre-training.

With validation loss as the objective, the next challenge is how to optimize it. Evaluating the objective for a given set of weights is computationally expensive, as it requires training a model from scratch on the weighted data set and evaluating the corresponding validation loss. Existing approaches tackle this in two broad ways. One line uses surrogate models to approximate the mapping from domain weights to model performance (Ye et al., 2024; Liu et al., 2024); fitting such models can still require large amounts of retrainings. For instance, in Liu et al. (2024), fitting the surrogate requires more than ten times as many training runs as there are domains. Another line performs local approximations, assuming only a single gradient step for the underlying model (Fan et al., 2023), which may not hold for practical learning rates.

Our work follows the surrogate-modeling line but differs by proposing a new parametric function to model performance versus domain weights, which can be reliably fit with only

about twice as many retraining runs as the number of domains. Crucially, existing methods often conduct this optimization at a smaller data scale to keep cost manageable, then directly apply the small-scale "best" mixture at large scale. We show that this scale-invariance assumption can break down in practice. To address this, we contribute a novel theoretical analysis that characterizes how the optimal domain ratio shifts across scales, enabling us to extend small-scale insights effectively to larger budgets.

Finally, the general notion that data curation should be scale-dependent has appeared in prior works (Sorscher et al., 2022; Goyal et al.). However, Sorscher et al. (2022) argues this point using a simplified analysis with a perception model—showing that larger scales favor "harder" samples while smaller scales favor "easier" samples—but does *not* propose a practical pipeline for scale-aware data selection. Our work addresses this gap by introducing a concrete method for scale-dependent curation at foundation-model scale. Meanwhile, Goyal et al. focuses on the CLIP model (Radford et al., 2021), finding that different training epochs call for different data-selection thresholds. By contrast, our setting involves (i) LLM pre-training (with typically one epoch), and (ii) different data modality and selection granularity.

## 3 Methodology

**Evidence of scale dependence.** A simple experiment illustrates that domain weighting is not one-size-fits-all: we derive two data mixes (with a procedure introduced later) and compare them at different training budgets. As shown in Figure 1, when tested at 0.3B tokens, Mix A beats Mix B as measured by validation perplexity reduction compared against uniform weights, but at 1.2 B tokens Mix B outperform Mix A. This flip indicates that a domain which helps more at a smaller scale may not remain a better choice at a larger scale, while another domain initially less impactful can become more valuable as training grows. Consequently, a scale-aware approach is needed so that domain weights adapt as more tokens are introduced.

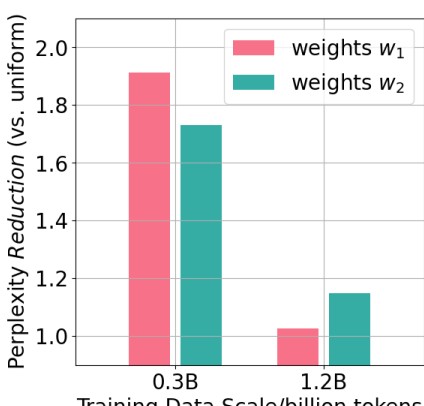

Figure 1: **Domain weights that excel at one scale may underperform at another.** Weights $w_1$ and $w_2$ are obtained by running DDO (as introduced in Section 3.2.1) at 0.3B and 1.2B, respectively.

### 3.1 Problem Formulation

To capture how domain importance shifts with the total training budget, we now formalize the scale-dependent domain reweighting problem. This framework lets us solve for a better domain mixture at any given data scale.

**Notations and setup.** We consider $m$ domains $\{D_1, \ldots, D_m\}$, each with a large pool of training examples. A *domain mix* is specified by a weight vector $\mathbf{w} = [w_1, \ldots, w_m]^\top$ on the probability simplex $\mathbb{W}^m := \left\{ \mathbf{w} \in \mathbb{R}^m \,\middle|\, \sum_{i=1}^m w_i = 1, w_i \geq 0 \text{ for all } i \right\}$. Given a total budget of $N$ tokens, let $N_i = \lfloor w_i \cdot N \rfloor$ be the number of tokens chosen from domain $D_i$. We denote this resulting dataset by $S(N, \mathbf{w}) = \{S_1, \ldots, S_m\}$, where $S_i \subseteq D_i, |S_i| = N_i$. Training a model $\boldsymbol{\theta}$ on $S(N, \mathbf{w})$ means solving an empirical risk minimization (ERM) objective: $\boldsymbol{\theta}^*(N, \mathbf{w}) = \arg\min_{\boldsymbol{\theta}} \mathcal{L}(\boldsymbol{\theta}, S(N, \mathbf{w}))$, where $\mathcal{L}$ is a next-token prediction loss.

**Objective function.** We assess a domain mix $\mathbf{w}$ by measuring the validation loss $\mathcal{L}^v(\boldsymbol{\theta}^*(N, \mathbf{w})) = \mathcal{L}(\boldsymbol{\theta}^*(N, \mathbf{w}), D^v)$ on a held-out dataset $D^v$. We then seek the mix $\mathbf{w}$ that *minimizes* this validation metric at scale $N$:

$$\mathbf{w}^* = \arg\min_{\mathbf{w} \in \mathbb{W}^m} \mathcal{L}^v(\boldsymbol{\theta}^*(N, \mathbf{w})). \tag{1}$$

Because $\boldsymbol{\theta}^*$ depends on $\mathbf{w}$ via ERM, this becomes a *bi-level optimization* problem. Since no closed-form expression exists for $\boldsymbol{\theta}^*$, any gradient-based approach must approximate

$\frac{\partial}{\partial \mathbf{w}} \mathcal{L}^v(\boldsymbol{\theta}^*(N, \mathbf{w}))$. Traditional bi-level methods rely on higher-order derivatives with respect to model parameters for this approximation, but such techniques become infeasible at the scale of modern foundation models (Liu et al., 2021).

## 3.2 Our Solution

We propose a **two-stage** framework, AutoScale, for finding scale-aware data compositions, which first approximates the optimal data mix at small scales and extrapolates to a larger target scale:

1. Direct Data Optimization (DDO): At smaller, computationally feasible scales, we learn a mapping from domain weights to validation loss. This reduces the original bi-level problem to a single-level convex optimization—letting us approximate the "best" domain mix for that smaller budget.

2. Optimal Mix Projection: Building on a theoretical analysis of how domain importance changes with total tokens, we then extrapolate those small-scale DDO solutions to a larger data budget.

### 3.2.1 Direct data optimization

Direct Data Optimization (DDO) is a practical method for approximating the solution to the bi-level domain-weighting problem at relatively small data scales. The key idea of DDO is to approximate the validation loss $\mathcal{L}^v(\boldsymbol{\theta}^*(N, \mathbf{w}))$ as a *parametric function* of the domain-weight vector $\mathbf{w}$. This effectively reduces our bi-level objective (choose $\mathbf{w}$ while also training $\theta$) to a *single-level* optimization, which can be solved efficiently via standard gradient-based methods.

We begin by noting that the validation loss can be *represented* by scaling laws as a function of data size for each individual domain. We *model* the dependence of validation loss on the size of data from each domain, then *aggregate* these functions to derive the final approximation for the validation loss on $\mathbf{w}$.

Drawing inspiration from neural scaling laws—which indicate a power-law relationship between training data scale and validation loss (Kaplan et al., 2020)—we assume that validation loss as a function of domain $i$'s data size follows

$$\mathcal{L}^v(\boldsymbol{\theta}^*(N, \mathbf{w})) \approx (N_0^i + w_i \cdot N)^{-\gamma_i} + \ell_i.$$

Here, $w_i$ denotes the fraction of the total token budget $N$ allocated to domain $i$. The term $N_0^i$ represents an the "equivalent data size" contributed by domains other than $i$, while $\gamma_i$ governs how quickly domain $i$ reaches a point of diminishing returns. Lastly, $\ell_i$ represents the irreducible term in the loss function.

To learn these parameters $\{N_0^i, \gamma_i, \ell_i\}$ for each domain $i$, we retrain the model after perturbing $w_i$ upward and downward, measure the change in total validation loss, and then fit $(N_0^i + w_i \cdot N)^{-\gamma_i} + \ell_i$ via least squares. Because $\mathcal{L}^v$ aggregates the effects of data size for each domain, our final approximation for the validation loss is:

$$\mathcal{L}^v(\boldsymbol{\theta}^*(N, \mathbf{w})) \approx \sum_{i=1}^{m} (N_0^i + w_i \cdot N)^{-\gamma_i} + \ell_i.$$

Once we have fitted the parameters, we can *directly optimize* over $\mathbf{w}$ subject to $\sum_{i=1}^{m} w_i = 1$ to approximate the optimal domain mix under the total token budget $N$.

Because DDO only requires retraining at $(2m + 1)$ mixes (one baseline plus up/down perturbations for each of the $m$ domains), it is far cheaper than a naive zero-order method that retrains the model at every weight update. Nevertheless, DDO is best suited for moderate domain counts ($m$) and data scales ($N$). For much larger target scales, we introduce a *second stage* that extrapolates the "best" DDO mix from smaller scales to significantly bigger budgets, all *without* additional retraining.

### 3.2.2 Optimal mix projection

Our method for extrapolating domain mixes to much larger training budgets hinges on a novel theoretical result that we developed to characterize how the optimal mix ratio depends on the total data scale. Note that all inverses $(\cdot)^{-1}$, products, and exponentiations on vectors below are understood *elementwise*.

---

**Theorem 1: Scale-Dependent Optimal Composition**

Consider the optimization problem

$$\min_{\mathbf{N}}\left\{\sum_{i=1}^{m}\beta_i N_i^{-\gamma_i} \;\Big|\; \sum_{i=1}^{m}N_i = N\right\},$$

where $\beta_i \geq 0$ and $\gamma_i \geq 0$ for all $i$, and $\mathbf{N} = (N_1,\ldots,N_m)$ denotes the domain allocations. Let $\mathbf{N}^*(N)$ be the *optimal allocation* that minimizes the sum above for a total budget $N$. For two distinct budgets $N^{(1)} \neq N^{(2)}$, and any larger budget $N^{(3)}$, suppose there is a constant $k > 0$ such that

$$\mathbf{N}(N^{(3)}) \;=\; \mathbf{N}^*(N^{(2)})\left[\left(\mathbf{N}^*(N^{(1)})\right)^{-1}\mathbf{N}^*(N^{(2)})\right]^k, \quad \text{with}\ \sum_{i=1}^{m}N_i(N^{(3)}) = N^{(3)}.$$

Then $\mathbf{N}(N^{(3)})$ is also the optimal allocation for the budget $N^{(3)}$, *i.e.*,

$$\mathbf{N}(N^{(3)}) \;=\; \arg\min_{\mathbf{N}}\left\{\sum_{i=1}^{m}\beta_i N_i^{-\gamma_i} \;\Big|\; \sum_{i=1}^{m}N_i = N^{(3)}\right\} \;=\; \mathbf{N}^*(N^{(3)}).$$

---

**Proof overview (high-level).**  At optimality, the first-order (KKT) conditions impose that each domain's partial derivative of the loss matches up to a single Lagrange multiplier. From this, we can derive how each domain's optimal allocation $N_i^*(N)$ scales when transitioning from budget $N^{(1)}$ to another budget $N^{(2)}$. These domain-by-domain scaling factors do not depend on the absolute size of $N$, only on the relative shifts between domains, which in turn yields an exponential-style expression for the optimal allocation at a third budget $N^{(3)}$. Thus, once we know the optimal allocations at two budgets, we can directly construct the optimal allocation for any larger budget *without* re-solving the entire optimization.

**Interpretation of the theory.**  The statement above assumes each domain $D_i$ contributes $\beta_i N_i^{-\gamma_i}$ *independently* to the total validation loss. In Appendix B.2, we generalize this to cases where domains may overlap by treating the evaluation as composed of multiple "latent skills" (Tiong et al.); the same exponential-style scaling behavior still emerges.

We defer the full proof to Appendix B.1 (where we employ first-order optimality/KKT conditions), but the key insight is that domains *saturate* at different rates depending on their exponents $\gamma_i$. Specifically, a domain $D_i$ with a small $\gamma_i$ saturates more slowly and thus continues to yield benefits at larger budgets, receiving an increasingly bigger fraction of tokens as $N$ grows. In contrast, a large $\gamma_i$ indicates that $D_i$ quickly saturates, so it is favored primarily at smaller scales.

Concretely, this means the "optimal mix ratio" is *not* constant across all scales. As the total budget $N$ increases, domains with smaller $\gamma_i$ are allocated a larger share. The theorem's exponential-style update precisely captures these changing allocations, enabling us to predict the best mix at a higher budget given the solutions at two smaller budgets—*without* re-solving the entire optimization problem.

### 3.2.3 Overall algorithm

Having established that the optimal domain mix varies predictably with training budget, we now summarize AutoScale , our proposed two-stage approach to optimize data mix.

**Stage 1** (pseudocode provided in Algorithm 1): Pick two feasible scales $N^{(1)}$ and $N^{(2)}$ (with $N^{(1)} < N^{(2)}$), where retraining the model is still affordable. Run DDO to obtain optimal allocations $\mathbf{N}^*(N^{(1)})$ and $\mathbf{N}^*(N^{(2)})$.

**Stage 2** (pseudocode provided in Algorithm 2): Leveraging our theoretical result, we automatically predict the optimal domain mix for any larger scale. Specifically, starting from the optimal domain allocation $\mathbf{N}^*\big(N^{(2)}\big)$, we repeatedly "scale up" by multiplying by $\left[(\mathbf{N}^*(N^{(1)}))^{-1}\,\mathbf{N}^*\big(N^{(2)}\big)\right]^{\delta}$. Each such update yields a new allocation at a larger budget than before. We continue until reaching or exceeding the target budget $N^{\text{tgt}}$. By adjusting the resolution $\delta$, we control the granularity of these updates, thus reaching the target scale with any desired accuracy.

## 4 Evaluation

Our evaluation aims to address the following questions:

- **(RQ1)** *Does DDO yield better domain weighting at smaller scales?* In our approach, DDO is designed to approximate the best data mix at a given scale. While we cannot verify its optimality, we want to see if DDO meaningfully improves domain weighting compared to baseline methods (Section 4.1).

- **(RQ2)** *Can AutoScale—DDO at smaller scales plus our theoretical projection to larger scales—achieve training efficiency and performance benefits when direct DDO at large scale is prohibitively expensive?* (Section 4.2)

**Overview of experimental settings.** We provide an overview here and defer the full details to Appendix C. **(I) Models and datasets.** We experiment with two architectures—GPT-2 Large (774M parameters) and BERT (110M), training on up to 10B tokens. While this budget is comparatively small for the latest LLM regimes, it already **exceeds** the data scales used in many existing domain-reweighting studies (Fan et al., 2023; Chen et al., 2024a), serving as a feasible testbed for *prototype* ideas in non-commercial settings. Specifically, we pre-train GPT-2 Large on the RedPajama dataset (Computer, 2023),which spans seven domains (e.g., Common Crawl, C4 (Raffel et al., 2020), GitHub, Wikipedia, ArXiv, StackExchange). For BERT, we use data from five sources—Amazon Reviews, Arxiv, Books, Wikipedia, and Open WebText (Gokaslan & Cohen, 2019). **(II) Baselines.** We compare our methods against several baseline strategies. **Uniform** samples data from each domain, leading to the same token count per domain. **Llama weights** are a curated set of heuristically tuned domain weights from the LLaMA-1/2 models (Touvron et al., 2023). **DoReMi** (Xie et al., 2024) is a seminal paper in this domain-reweighting space, offering an early, principled approach to finding domain weights. **Data mixing law** (Ye et al., 2024) and **RegMix** (Liu et al., 2024) represent the latest state-of-the-art. **(III) Metrics.** We measure *test* perplexity and also evaluate downstream performance to confirm that improvements extend to practical tasks.

### 4.1 Evaluating DDO

**Effectiveness of DDO-optimized weights.** We perform DDO on GPT-2 Large at two different data scales (0.3B and 1.2B tokens) to obtain DDO weights specifically optimized for each scale. We then retrain the model under these DDO-derived weights and compare the evaluation loss against two baselines: (1) Uniform (no reweighting), and (2) RegMix, the latest state-of-the-art approach. (We omit DoReMi here, as it has been surpassed by RegMix (Liu et al., 2024).) For each set of weights, the model is trained at both 0.3B and 1.2B tokens, with results in Table 1. At both scales, DDO-optimized domain weights significantly outperform the Uniform baseline, achieving a notably lower evaluation loss. DDO-optimized weights also surpass RegMix when models are trained at the same scale as the domain-weight optimization, indicating that DDO finds more effective domain weights than RegMix. Further, RegMix does not consider adaptation for training models at different scales. Applying RegMix optimized weights on larger data scales appears less effective,

evident by the *widening gap* between its performance from DDO's. Notably, the DDO-derived weights yield the strongest gains at the scale for which they were optimized, while showing less advantage when used at a *different* scale, highlighting the *scale-dependent* nature of domain weighting.

| Weights/Actual training scale | 0.3B training tokens | 1.2B training tokens |
|---|---|---|
| Uniform Weights | 48.04 | 28.11 |
| RegMix Weights (optimized at 0.3B) | 46.56 (-1.48) | 27.86 (-0.25) |
| DDO Weights (optimized at 0.3B) | **46.13 (-1.91)** | 27.09 (-1.02) |
| DDO Weights (optimized at 1.2B) | 46.31(-1.73) | **26.97 (-1.14)** |

Table 1: GPT-2 Large trained with DDO optimized domain weights achieve significantly reduced test perplexity compared to with non-optimized, uniform weights, also outperforming RegMix. DDO optimized weights appear most performant at the data scale they were optimized.

In addition, we apply DDO to BERT at 0.3B tokens; the resulting model performance from the DDO-optimized weights is shown in Fig. 2. These weights reduce the model's validation loss on all training domains *and* on held-out non-training domains, demonstrating DDO's effectiveness in improving training efficiency. Furthermore, when evaluated on the GLUE benchmark and the SQuAD dataset, the DDO-optimized weights also yield a notable improvement in downstream task performance.

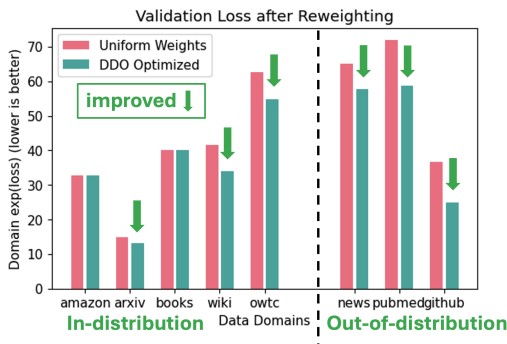
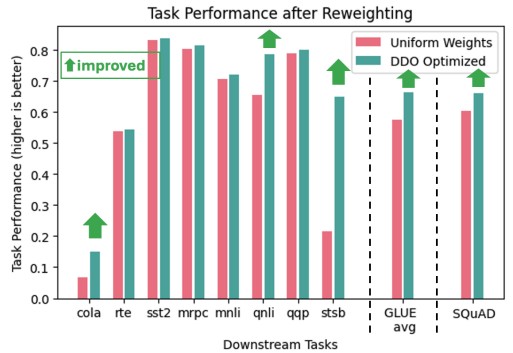

(a) Validation Loss (↓ lower is better)  (b) Task Performance (↑ higher is better)

Figure 2: Optimizing domain weights with DDO algorithm for pre-training Encoder-only LMs (BERT). DDO substantially reduces validation loss. After reweighting, all training domains' loss has decreased or remained unchanged. Out-of-domain loss on non-training domains also decreased considerably. Enhanced performance is observed on all GLUE tasks (eval metric: cola: Matt. corr., stsb: Pearson corr., rest: acc.) and SQuAD (acc.).

**Analyzing DDO's effectiveness.** Recall that the key idea of DDO is to use a power-law–based parametric function to predict validation loss from domain weights. A major factor in DDO's effectiveness lies in the *accuracy* of this function. We quantify its predictive power via the *average absolute relative error (AAR)* between the predicted and actual losses. In our experiments, the AAR is 1.00%, indicating that DDO's modeling closely reflects actual loss.

## 4.2 Evaluating AutoScale

**Effectiveness of our extrapolated weights.** Recall that AutoScale is a two-stage pipeline: first, run DDO at smaller scales to identify domain weights, then extrapolate those weights to a larger scale. We call the resulting allocation the *AutoScale weights*. For GPT-2 Large, we run DDO on up to 0.6B tokens, then extrapolate to 3B and 10B tokens. Figure 3 shows the change of test perplexity during training for models trained with 10B tokens using AutoScale weights versus baseline allocations. AutoScale consistently outperforms every baseline by a 28–38% margin and also demonstrates advantageous downstream performance. Table 2 demonstrates the results on 3B tokens, revealing that AutoScale maintains its superiority in both final loss achieved and faster convergence. Table 3 examines domain-wise test perplexities; AutoScale weights significantly reduce the loss on the Books domain and

improve worst-domain perplexity, also yielding a better average across domains. Finally, Table 4 evaluates eight downstream tasks. The model trained with AutoScale weights achieves the best overall performance, further underscoring the effectiveness of our extrapolated domain weights.

For BERT, we train up to 288k steps (approximately 120% of the original BERT-base budget (Devlin et al., 2018)). Table 10 shows that, compared to uniform (no reweighting), AutoScale yields a 16.7% speed-up at most data scales and a 10% speed-up at the largest scale, demonstrating consistent effectiveness. However, these gains are smaller than those observed for GPT-2 Large, indicating that different architectures and training objectives may respond differently to domain reweighting. This is also hinted at in Figure 10, where the evaluation loss shows a more uniform response to each domain, suggesting fewer benefits from reweighting in BERT's setup.

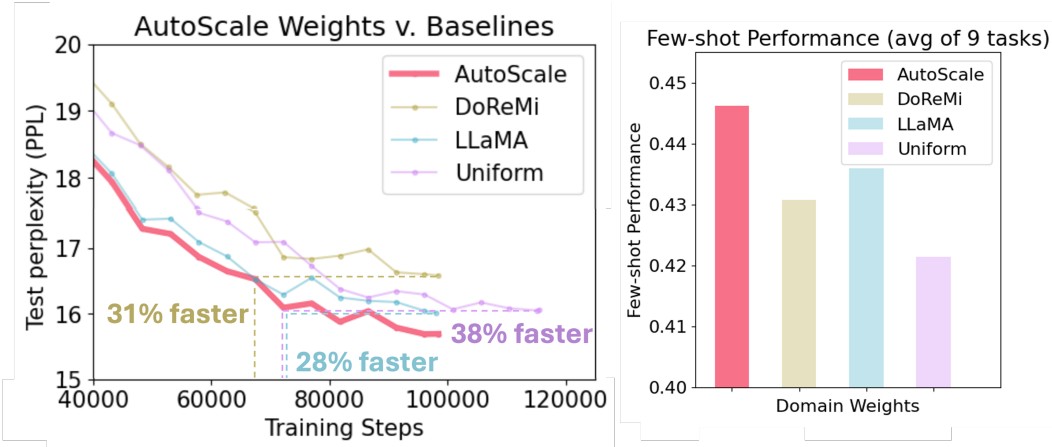

Figure 3: Training 774M Decoder-only LMs (GPT-2 Large) for 10B tokens (96k steps). AutoScale-predicted domain weights decrease test perplexity at least 28% faster than any baseline with up to 38% speed up, achieving best overall task performance.

| Weights (3B training tokens) | Final Perplexity (PPL) | AutoScale Speed Improvement (% steps saved to final PPL) |
|---|---|---|
| AutoScale (ours) | **21.123** | - |
| DoReMi | 21.676 | 25% |
| Data Mixing Laws | 23.333 | 37% |
| LLaMA | 22.944 | 31% |
| RegMix | 21.740 | 28% |
| Uniform (30% more tokens) | 21.736 | 37% |

Table 2: Domain perplexity for 774M Decoder-only LMs (GPT-2 Large) trained for 3B tokens. AutoScale -predicted weights decreases val loss at least 25% faster than any baseline with up to 37% speed up. Despite LLaMa weights being very different from uniform weights, they yield highly similar training efficiency at these data scales.

| Domain/Method | AutoScale (ours) | DoReMi | Data Mixing Laws | LLaMA | RegMix | Uniform (30% more tokens) |
|---|---|---|---|---|---|---|
| Common Crawl | 25.598 | **24.116** | 30.824 | 21.464 | 24.430 | 28.351 |
| Github | 7.482 | 6.678 | **5.845** | 7.376 | 6.145 | 5.784 |
| Books | **29.162** | 33.324 | 34.450 | 35.533 | 32.985 | 31.140 |
| Wikipedia | 18.828 | **17.154** | 26.795 | 21.110 | 20.177 | 19.570 |
| C4 | **34.242** | 39.429 | 38.521 | 37.393 | 39.654 | 40.323 |
| Stack Exchange | 15.991 | 15.393 | **14.519** | 20.133 | 15.225 | 13.890 |
| Arxiv | 16.558 | 15.638 | **12.372** | 17.598 | 13.563 | 13.082 |
| Average | **21.123** | 21.676 | 23.333 | 22.944 | 21.740 | 21.736 |
| Worst-domain | **34.242** | 39.429 | 38.521 | 37.393 | 39.654 | 40.323 |

Table 3: Domain perplexity for 774M GPT-2 Large trained for 3B tokens. AutoScale notably achieves the lowest average test perplexity while also significantly decreasing worse-domain perplexity.

**Examining how domain importance evolves with scale.** To illustrate the shift in domain importance, we first run DDO on GPT-2 Large across scales ranging from 30M to 1.2B tokens.

| Method/Task | Avg | pubmedqa | piqa | hellaswag (10-shot) | crows_pairs _english | boolq | arc_easy | truthfulqa _mc2 | hellaswag (zero-shot) |
|---|---|---|---|---|---|---|---|---|---|
| **AutoScale (ours)** | **0.4746** | **0.536** | **0.6202** | **0.3021** | 0.5850 | 0.6141 | **0.3977** | 0.4385 | **0.3030** |
| Uniform Weights | 0.4514 | 0.438 | 0.6115 | 0.2923 | 0.5886 | 0.5636 | 0.3742 | 0.4526 | 0.2907 |
| LLaMA Weights | 0.4585 | 0.492 | 0.6055 | 0.2944 | **0.5903** | 0.5612 | 0.3956 | 0.434 | 0.2952 |
| Data Mixing Laws | 0.4610 | 0.468 | 0.6061 | 0.2951 | 0.5778 | **0.6162** | 0.3771 | **0.4537** | 0.2938 |
| DoReMi | 0.4482 | 0.468 | 0.5985 | 0.2886 | 0.5742 | 0.5410 | 0.3750 | 0.4505 | 0.2896 |
| RegMix | 0.4642 | 0.526 | 0.6077 | 0.2907 | 0.5850 | 0.6000 | 0.3721 | 0.4455 | 0.2868 |

Table 4: Task performance for 774M GPT-2 Large trained for 3B tokens. Models trained with AutoScale -predicted weights achieve the best overall performance across the tasks.

| Data Scale/steps | 18k | 36k | 72k | 144k | 288k |
|---|---|---|---|---|---|
| Final Loss (exp) | 38.32 | 16.94 | 10.97 | 8.13 | 6.30 |
| Steps Saved | 5k (28%) | 5k (14%) | 10k (14%) | 20k (14%) | 20k (10%) |

Table 5: AutoScale notably improving training efficiency for BERT models on all scales–even for a considerably large scale, 288k steps, the speedup margin remains visible.

Figure 4(a) shows that the DDO-optimized weights differ *visibly* at each scale, highlighting a clear shifting pattern. Data sources with more standardized formats (Wikipedia, scientific papers)—often regarded as "high quality"—dominate at smaller scales but exhibit sharp diminishing returns as the data budget grows. By contrast, domains with more diverse examples (C4, CommonCrawl) continue to lower training loss even at higher scales.

Consistently, taking DDO-optimized weights from up to 0.6B tokens, we use our theory to project how the composition would shift at scales beyond 1.2B. Figures 4bd) and 6 show that as the training data scale grows, diverse domains (C4, CommonCrawl) command a larger share of the mix compared to "standard" domains. We observe a similar pattern with BERT, where we extrapolate the DDO-optimized weights at 0.5B tokens to even larger scales, revealing that domains like WebText and Amazon Reviews gain significance over clean, standardized data (Wikipedia, Arxiv) (see Fig. 11). A plausible explanation is that "diverse" data provides broader topical coverage and linguistic styles, while "standard" data saturates more quickly.

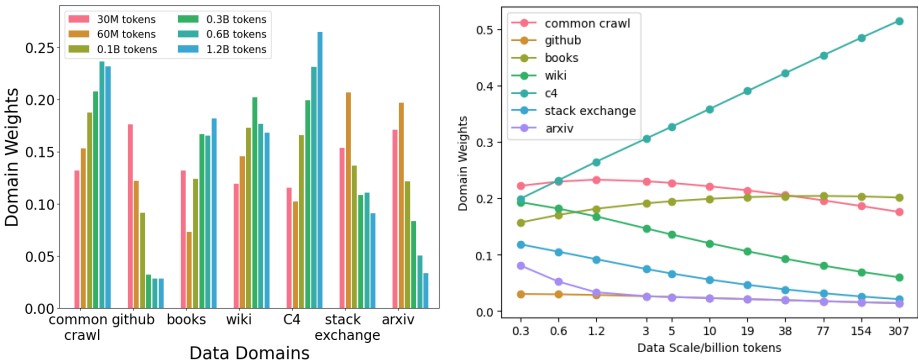

(a) DDO optimized domain weights.    (b) AutoScale projected domain weights.

Figure 4: Domain importance evolves with training data scales. (GPT-2 Large)

Note that these trends show *how* our approach predicts domain importance may evolve, not a proof that each extrapolated mix *guarantees* the best performance at its target scale. Nonetheless, the consistent shifting patterns across GPT-2 Large and BERT reinforce the idea that domain importance is *scale-dependent*.

## 5 Conclusions

This paper explores how the importance of each training domain shifts across different scales and proposes a scale-aware framework (AutoScale) that outperforms existing approaches across various architectures, datasets, and training scales. Still, our experimental settings remain limited in scale and the diversity of evaluations. Extending this work to larger

training budgets, additional data modalities, and broader benchmarks would clarify how well our insights generalize beyond the current scope. Another exciting next step is to adapt AutoScale for directly optimizing downstream metrics, moving beyond perplexity as a rough proxy for language-model quality.

# 6 Discussions

**Online vs. offline data mixing methods.** These represent two independent paths to approach data mixing problems in LM pre-training. Current industrial practices are typically offline. Offline methods, such as DoReMi (Xie et al., 2024), Doge (Fan et al., 2023), RegMix (Liu et al., 2024), DML (Data Mixing Laws, Ye et al. (2024)), and our proposed AutoScale, decouple data curation from the final training process. *This separation allows for the distribution of model development workloads across different teams (e.g., data, model, and training) working in parallel.* This approach is common in industrial settings for developing state-of-the-art LMs (e.g., Phi-4 (Abdin et al., 2024), Llama4 (Meta, 2025)). Notably, many offline methods like DoReMi, DML, and BiMIX (Ge et al., 2024) originate from industrial research, whereas current online methods (ODM (Albalak et al., 2023), Skill-it (Chen et al., 2024a), Aioli (Chen et al., 2024b), ADO (Adaptive Data Optimization, Jiang et al. (2024))) are primarily academic contributions.

Offline methods treat the training process integrally, primarily adjusting domain weights and training budgets before training commences. In contrast, online methods (e.g., ODM, Skill-it, Aioli, ADO) adjust domain weights dynamically during the training process. ODM, for instance, models online domain weighting as a bandit problem and updates domain weights based on local gradients. Considering the learning rate often changes throughout training, the loss landscape may be non-convex where a greedy approach cannot guarantee global optimality. Skill-It primarily considers fine-tuning scenarios, while Aioli extends this to small-scale LM pre-training. Aioli performs online estimation of the scaling law relationship between each pair of training and validation domains (in contrast to DML's offline estimation), while ADO considers scaling laws for individual training domains. The scaling law relationships these methods were originally proposed for adjusting training data size within a fixed training pipeline (Kaplan et al., 2020; Hoffmann et al., 2022). They may not always accurately track a training process and its dynamics, especially with a varying learning rate, complicating their conceptual grounding. This creates space for future research contributions. For deployment in production pipelines, a better understanding on the predictability of the outcome under different training paradigms is desirable.

**Effect of repeated documents and "data-constrained setting"** The data constraint is relevant for real-world LM pre-training, particularly for low-resource domains. Beyond constraints on domain weights, practical limitations include the amount of available data and the maximum number of repetitions tolerated before significant performance degradation. Existing research on domain weighting has not explicitly modeled this factor. Despite this research gap, Muennighoff et al. (2023) offers positive evidence that current paradigms can still be effective under such constraints. The paper indicates that repeating data for up to 3 epochs has almost no negative impact, and models can still train reasonably well with up to 7 epochs of repetition. This suggests that in practice, repeating data from some domains a few times can be acceptable without significant detriment to performance. Current data mixing methods may be applied to these scenarios without major modifications.

**Data mixing for model-size scaling** Applying the data-scaling methodology to model-size scaling appears plausible. Scaling law papers (Kaplan et al., 2020; Hoffmann et al., 2022) reported similar scaling trends between validation loss and training data budget/model size. (Shukor et al., 2025) proposes to model the combined effects of data and model-size scaling as a linear addition and validates its effectiveness in optimizing domain weights for LLM, native multimodal model (NMM), and large vision models (LVM) pretraining.

Additional discussions on **Intuition for Stage 2 of** AutoScale, **Optimal mix projection** and **Number of predictors (scaling law components)** can be found in Appendix D.

## Impact Statement

Reducing the complexity and resource requirements associated with pretraining LLMs, AutoScale contributes to the democratization of AI. Smaller organizations, academic institutions, and individual researchers can more easily participate in cutting-edge AI research and development, fostering innovation and collaboration across the AI community. Moreover, learning from massive amounts of data requires large and costly computational resources, which not only consume substantial energy but also generate a significant carbon footprint, contributing to environmental issues. Furthermore, these resources quickly become obsolete due to the rapid pace of technological advancements, leading to e-waste. This research makes contributions to mitigating these issues by improving the efficiency of resource utilization in AI training.

## Acknowledgement

This work is supported in part by the National Science Foundation under grants IIS-2312794, IIS2313130, OAC-2239622, Amazon-Virginia Tech Initiative in Efficient and Robust Machine Learning, AWS computational credits, and the Commonwealth Cyber Initiative. The authors are grateful for Ankit Battawar and Alix Delgado from AWS, whose dedicated help and support were crucial for securing computing resources and implementing empirical studies.

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

# Appendices

# A  Algorithms and Operational Pipeline

---

**Algorithm 1** Direct Data Optimization (DDO)

---

**Require:** $m$ domains (data sources) with data $D_1 \ldots D_m$, data budget $N_0$ ($\ll$ for full-scale training), training dataset $S$, model parameters $\theta$, validation loss $\mathcal{L}_v$, perturbation ratio $r > 1$ (e.g., $r = 3$).

Initialize weights for all domains $\forall i \in \{1, \ldots m\}$: $w_i \leftarrow 1/m$;

Initialize training data for all domains $\forall i \in \{1, \ldots m\}$: sample $S_i \subset D_i$ where $|S_i| = w_i \cdot N$;

Train the model on data $S = \{S_1 \ldots S_m\}$ and evaluate its loss $\mathcal{L}_v^0 \leftarrow \mathcal{L}_v(\theta^*(S))$;

**for** $j$ from 1 to $m$ **do**

    $w_j^+ \leftarrow r \cdot w_j$;                                        ▷ Perturb domain weights (+)

    Resample $S_j^+ \subset D_j$ where $|S_j^+| = w_j^+ \cdot N$;

    Train the model on data $S = (\{S_1 \ldots S_m\} \setminus S_j) \cup S_j^+$ and evaluate its loss $\mathcal{L}_j^+ \leftarrow \mathcal{L}_v(\theta^*(S))$;

    $w_j^- \leftarrow \frac{1}{r} \cdot w_j$;                                       ▷ Perturb domain weights (-)

    Resample $S_j^- \subset D_j$ where $|S_j^-| = w_j^- \cdot N$;

    Train the model on data $S = (\{S_1 \ldots S_m\} \setminus S_j) \cup S_j^-$ and evaluate its loss $\mathcal{L}_j^- \leftarrow \mathcal{L}_v(\theta^*(S))$;

    OLS fit for scaling functions $N_0^i, \gamma_i, \ell_i = \arg\min_{N_0^i, \gamma_i, \ell_i} [\mathcal{L}_v^0 - (N_0^i + N_i)^{-\gamma_i} - \ell_i]^2 + [\mathcal{L}_{(+i)} - (N_0^i + N_i^+)^{-\gamma_i} - \ell_i]^2 + [\mathcal{L}_{(-i)} - (N_0^i + N_i^-)^{-\gamma_i} - \ell_i]^2$;

**end for**

Output optimized domain weights $\mathbf{w}^* = \arg\min_{\mathbf{w}' \in \mathbb{W}^m} \sum_{i=1}^m (N_0^i + w_i' \cdot N)^{-\gamma_i}$.

---

---

**Algorithm 2** AutoScale

---

**Require:** Optimal domain weights (obtained from DDO) $\mathbf{w}^{(1)*}$ at data scale $N^{(1)}$ and $\mathbf{w}^{(2)*}$ at data scale $N^{(2)}$, target data scale $N^{(t)}$, where $N^{(1)} < N^{(2)} < N^{(t)}$; resolution $\delta$.

Optimal domain data $\mathbf{N}^*(N^{(1)}) \leftarrow \mathbf{w}^{(1)*} \cdot N^{(1)}$;

Optimal domain data $\mathbf{N}^*(N^{(2)}) \leftarrow \mathbf{w}^{(2)*} \cdot N^{(2)}$;

Current data budget $N \leftarrow \sum_i N_i^{(2)*}$;

Optimal domain data under current data budget $\mathbf{N}^*(N) \leftarrow \mathbf{N}^*(N^{(2)})$;

**while** $N < N^{(t)}$ **do**

    Compute optimal domain data under the next data budget: $\mathbf{N}^*(N^{\text{next}}) \leftarrow \mathbf{N}^*(N)[(\mathbf{N}^*(N^{(1)}))^{-1}\mathbf{N}^*(N^{(2)})]^\delta$;

    Compute the next data budget $N^{\text{next}} \leftarrow \sum_i N_i^*$;

    Update current data budget $N \leftarrow N^{\text{next}}$;

**end while**

Output predicted optimal domain weights: $\hat{\mathbf{w}}^{(t)*} \leftarrow \mathbf{N}^*(N)/N$.

---

**Operational Pipeline (DDO)**

1. Train a base proxy model with uniform weights (or reference weights, if available);

2. At each time, add/reduce data quantity for one domain and re-train the proxy model;

3. Fit power law scaling functions and solve the optimization problem;

4. Iterate the process if necessary.

**Operational Pipeline (**AutoScale **)**

1. For two smaller training data scales $N^{(1)}$ and $N^{(2)}$ where re-training the model is affordable, find their corresponding optimal training data compositions $\mathbf{N}^*(N^{(1)})$ and $\mathbf{N}^*(N^{(2)})$ using DDO Algorithm described above;

2. Initialize current data budget at $N = N^{(2)}$;

3. With the chosen resolution $\delta$, predict the next optimal training data composition as $\mathbf{N}^*(N^{\text{next}}) = \mathbf{N}^*(N)[(\mathbf{N}^*(N^{(1)}))^{-1}\mathbf{N}^*(N^{(2)})]^\delta$, yielding optimal domain weights $w_i^* = N_i^*(N^{\text{next}})/N^{\text{next}}$ at new training data scale $N^{\text{next}} = \sum_i N_i^*(N^{\text{next}})$;

4. Update current data budget to $N = N^{\text{next}}$. Repeat this process until the target training data scale is reached.

## B Proofs for Section 3.2.2, Optimal mix projection

### B.1 Theorem 1: Scale-Dependent Optimal Composition

**Theorem B.1** (Scaling Law for Optimal Data Compositions **(restated)**). *Consider the following optimization problem*

$$\min_{\mathbf{N}} \left\{ \sum_{i=1}^{m} \beta_i N_i^{-\gamma_i} \,\middle|\, \sum_{i=1}^{m} N_i = N \right\}.$$

*For any two compute budgets $N^{(1)} \neq N^{(2)}$, let $\mathbf{N}^*(N^{(1)})$ and $\mathbf{N}^*(N^{(2)})$ be their respective minimizers. For any third data composition $\mathbf{N}(N^{(3)})$, if there exists some constant $k \in \mathbb{R}^+$ such that*

$$\mathbf{N}(N^{(3)}) = \mathbf{N}^*(N^{(2)})[(\mathbf{N}^*(N^{(1)}))^{-1}\mathbf{N}^*(N^{(2)})]^k,$$

*then, $\mathbf{N}(N^{(3)})$ is the minimizer for data budget $N^{(3)} = \sum_{i=1}^{m} N_i^{(3)}$, given as*

$$\mathbf{N}(N^{(3)}) = \arg\min_{\mathbf{N}} \left\{ \sum_{i=1}^{m} \beta_i N_i^{-\gamma_i} \,\middle|\, \sum_{i=1}^{m} N_i = N^{(3)} \right\} = \mathbf{N}^*(N^{(3)}).$$

*Proof.* **Setup:** We begin with the following optimization problem, defined at a given total training data scale $N^{(1)}$:

$$\min_{\mathbf{N}} \left\{ \sum_{i=1}^{m} \beta_i N_i^{-\gamma_i} \,\middle|\, \sum_{i=1}^{m} N_i = N^{(1)} \right\}.$$

Here, $\mathbf{N} = \text{diag}\{N_1, N_2, \ldots, N_m\}$ is a diagonal matrix whose diagonal entries are the amounts of data allocated to each of the $m$ domains.

For this problem, there exists a unique optimal solution $\mathbf{N}^*(N^{(1)}) = \text{diag}\{N_1^{(1)*}, N_2^{(1)*}, \ldots, N_m^{(1)*}\}$. This $\mathbf{N}^*(N^{(1)})$ represents the compute-optimal data composition at the data scale $N^{(1)}$.

**First-Order Conditions (KKT):** At optimality, the Karush–Kuhn–Tucker (KKT) conditions ensure that the partial derivatives of the objective function with respect to each $N_i$ are equal (up to the same Lagrange multiplier for the equality constraint $\sum_i N_i = N^{(1)}$). For any pair of domains $a$ and $b$, we must have:

$$\frac{\partial}{\partial N_a} \left( \sum_{i=1}^{m} \beta_i N_i^{-\gamma_i} \right) \bigg|_{N_a = N_a^{(1)*}} = \frac{\partial}{\partial N_b} \left( \sum_{i=1}^{m} \beta_i N_i^{-\gamma_i} \right) \bigg|_{N_b = N_b^{(1)*}}.$$

Computing these derivatives, we get:

$$-\beta_a \gamma_a (N_a^{(1)*})^{-\gamma_a - 1} = -\beta_b \gamma_b (N_b^{(1)*})^{-\gamma_b - 1}.$$

From this equality:

$$\frac{\beta_a \gamma_a}{\beta_b \gamma_b} = \frac{(N_a^{(1)*})^{\gamma_a + 1}}{(N_b^{(1)*})^{\gamma_b + 1}}.$$

Rearranging, we obtain a fundamental scaling relationship:

$$N_a^{(1)*} = \left( \frac{\beta_a \gamma_a}{\beta_b \gamma_b} (N_b^{(1)*})^{\gamma_b + 1} \right)^{\frac{1}{\gamma_a + 1}}.$$

**Scaling to a Second Data Scale $N^{(2)}$:** Now consider a different total data scale $N^{(2)} \neq N^{(1)}$, with the corresponding compute-optimal solution $\mathbf{N}^*(N^{(2)}) = \text{diag}\{N_1^{(2)*}, N_2^{(2)*}, \ldots, N_m^{(2)*}\}$.

Suppose we know how the optimal amount of data for domain $b$ changes from $N^{(1)}$ to $N^{(2)}$. Specifically, let:

$$N_b^{(2)*} = m \cdot N_b^{(1)*}$$

for some scaling factor $m > 0$.

Applying the same relationship used for the first scale, but now at the second scale, we find that for domain $a$:

$$N_a^{(2)*} = \left( \frac{\beta_a \gamma_a}{\beta_b \gamma_b} (N_b^{(2)*})^{\gamma_b+1} \right)^{\frac{1}{\gamma_a+1}} = \left( \frac{\beta_a \gamma_a}{\beta_b \gamma_b} (m \cdot N_b^{(1)*})^{\gamma_b+1} \right)^{\frac{1}{\gamma_a+1}}.$$

This simplifies to:

$$N_a^{(2)*} = m^{\frac{\gamma_b+1}{\gamma_a+1}} N_a^{(1)*}. \tag{2}$$

Notice that $m^{\frac{\gamma_b+1}{\gamma_a+1}} \neq m$ in general. Thus, when the budget scales by a factor $m$ in domain $b$, the optimal amount for domain $a$ scales by a different factor. This shows that the optimal composition is scale-dependent.

**Predicting a Third Scale** $N^{(3)}$**:** We now know the optimal compositions at two scales $N^{(1)}$ and $N^{(2)}$. Consider a third scale $N^{(3)}$ and its optimal solution $\mathbf{N}^*(N^{(3)}) = \mathrm{diag}\{N_1^{(3)*}, N_2^{(3)*}, \ldots, N_m^{(3)*}\}$.

If we choose $N_b^{(3)*}$ such that:

$$\frac{N_b^{(3)*}}{N_b^{(2)*}} = \frac{N_b^{(2)*}}{N_b^{(1)*}}, \tag{3}$$

then the change in $N_b$ from $N^{(2)}$ to $N^{(3)}$ mirrors the change from $N^{(1)}$ to $N^{(2)}$.

Since the scaling exponent $\frac{\gamma_b+1}{\gamma_a+1}$ remains the same, this symmetrical setup leads to:

$$N_a^{(3)*} = \frac{(N_a^{(2)*})^2}{N_a^{(1)*}}.$$

**Matrix Form:** Because all domains scale in a similar fashion, we can write this relationship compactly using diagonal matrices. Define:

$$\mathbf{N}^*(N^{(i)}) = \mathrm{diag}\{N_1^{(i)*}, N_2^{(i)*}, \ldots, N_m^{(i)*}\}.$$

The element-wise relationship $\frac{(N_a^{(2)*})^2}{N_a^{(1)*}}$ for each domain $a$ can be expressed as:

$$\mathbf{N}^*(N^{(3)}) = \mathbf{N}^*(N^{(2)})(\mathbf{N}^*(N^{(1)}))^{-1}\mathbf{N}^*(N^{(2)}).$$

Here, $(\mathbf{N}^*(N^{(1)}))^{-1}$ is the inverse of the diagonal matrix $\mathbf{N}^*(N^{(1)})$, obtained by taking the reciprocal of each diagonal element $N_a^{(1)*}$.

We have shown that given two distinct data scales $N^{(1)}$ and $N^{(2)}$ and their corresponding optimal solutions $\mathbf{N}^*(N^{(1)})$ and $\mathbf{N}^*(N^{(2)})$, one can construct a third optimal solution $\mathbf{N}^*(N^{(3)})$ using the formula:

$$\mathbf{N}^*(N^{(3)}) = \mathbf{N}^*(N^{(2)})(\mathbf{N}^*(N^{(1)}))^{-1}\mathbf{N}^*(N^{(2)}).$$

This relationship holds without needing to explicitly estimate the parameters $\gamma_i$ or $\beta_i$, and it confirms that the optimal data composition is scale-dependent. Thus, the given scaling law for optimal data compositions is established.

**Generalization to Prediction for Any Data Scale:** Finally, we generalize from the case in Eq. (3) to allow prediction of optimal data composition for *any* data scale. Consider for some constant $\forall k \in \mathbb{R}^+$, we choose $N_b^{(k)*}$ such that

$$\frac{N_b^{(k)*}}{N_b^{(2)*}} = \left( \frac{N_b^{(2)*}}{N_b^{(1)*}} \right)^k = m^k,$$

Same as the procedure in Eq. (2), KKT optimality conditions yield the corresponding optimal data quantity for domain a at the same scale as $N_b^{(k)*}$ as

$$N_a^{(k)*} = (m^k)^{\frac{\gamma_b+1}{\gamma_a+1}} N_a^{(2)*} = \left( \frac{N_a^{(2)*}}{N_a^{(1)*}} \right)^k N_a^{(2)*}.$$

Rearranging in the matrix form, we have the following formula

$$\mathbf{N}^*(N^{(k)}) = \mathbf{N}^*(N^{(2)})[\mathbf{N}^*(N^{(2)})(\mathbf{N}^*(N^{(1)}))^{-1}]^k,$$

which concludes the proof.

**Application in AUTOSCALE:** Note that this formulate holds for any $k \in \mathbb{R}^+$. Thus, by scanning through the values of $k$, one can find optimal data composition $\mathbf{N}^*$ for any target data scale $N = \sum_i N_i^*$. *In practice, for a target data scale $N > N^{(2)} = \sum_i N_i^{(2)*}$, one only needs to conduct a line search along $k > 1$ to find the value of $k$ where $\sum_i N_i^{(k)*} = N$ to determine its corresponding optimal data composition $\mathbf{N}^*$.* $\qquad\square$

*Remark* B.2 (An example). This example helps visualize the operation pipeline.

If at training data scale $N^{(1)} = N_a^{(1)} + N_b^{(1)} = 200$, we have optimal domain data composition as $N_a^{(1)*} = 100, N_b^{(1)*} = 100$ (50% − 50%); and at scale $N^{(2)} = N_a^{(2)} + N_b^{(2)} = 500$, we have optimal domain data composition as $N_a^{(2)*} = 300, N_b^{(2)*} = 200$ (60% − 40%). Then, from the theorem, when the optimal domain data composition has $N_a^{(3)*} = (N_a^{(2)*})^2 / N_a^{(1)*} = 900$, we can predict $N_b^{(3)*} = (N_b^{(2)*})^2 / N_b^{(1)*} = 400$, which gives the optimal ratio at $N^{(3)} = N_a^{(3)} + N_b^{(3)} = 1300$ as 69% − 31%.

Similarly,

For $N_a^{(4)*} = 2700$, we have $N_b^{(4)*} = 800$, which gives the optimal ratio at $N^{(4)} = 3500$ as 77% − 23%

For $N_a^{(5)*} = 8100$, we have $N_b^{(5)*} = 1600$, which gives the optimal ratio at $N^{(5)} = 9700$ as 84% − 16%

For $N_a^{(6)} = 24300$, we have $N_b^{(6)*} = 3200$, which gives the optimal ratio at $N^{(6)} = 27500$ as 88% − 12%

For $N_a^{(7)*} = 72900$, we have $N_b^{(7)*} = 6400$, which gives the optimal ratio at $N^{(7)} = 79300$ as 92% − 8%

For $N_a^{(8)*} = 218700$, we have $N_b^{(8)*} = 12800$, which gives the optimal ratio at $N^{(8)} = 231500$ as 94% − 6%

For $N_a^{(9)*} = 656100$, we have $N_b^{(9)*} = 25600$, which gives the optimal ratio at $N^{(9)} = 681700$ as 96% − 4%

We visualize it in Fig. 5.

## B.2 Scaling Latent Skills

We extend this theory to a general case where the evaluation loss is the perplexity averaged over training domains. Consider the evaluation is composed of a number of *independent* sub-tasks ("latent skills" (Tiong et al.)) which are hidden variables, where each of them observes a power law scaling law relationship with the amount of data contributing to this task ("equivalent data size"), $\mathcal{L} = \ell_0 + \beta_a \cdot K_a^{-\gamma_a} + \beta_b \cdot K_b^{-\gamma_b} + \beta_c \cdot K_c^{-\gamma_c} + \cdots$ where

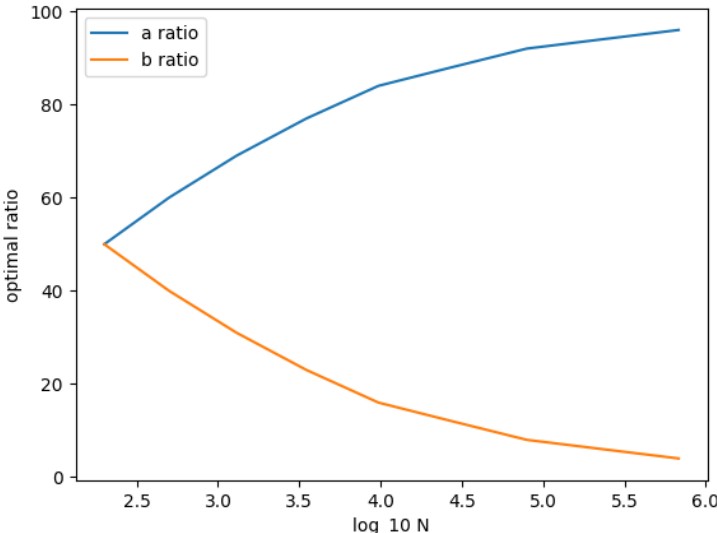

Figure 5: Illustration: optimal data composition scales in exponential-style functions with training data quantity.

scalar $K_j \geq 0$ denote equivalent data size for $skill_j$, and constants $(\beta_j, \gamma_j) \geq 0$ are coefficients associated with $skill_j$, respectively. Mathematically, these latent skills can be seen as an orthogonal basis that spans the space of evaluation loss.

Consider training data from each domain $D_i$ contributes to these skills to varying degrees, where Equivalent data size for $skill_j$, $K_j$, is given as $K_j = c_{j,1} \cdot N_1 + c_{j,2} \cdot N_2 + \cdots$ where $N_i = w_i \cdot N$ denotes the amount of training data from domain $D_i$ and constant $c_{j,i}$ is the coefficient measuring the degree of contribution between domain $D_i$ and $skill_j$. Defining diagonal matrices for training data composition $\mathbf{N} = diag\{N_1, N_2, \cdots\}$ and skill data composition $\mathbf{K} = diag\{K_a, K_b, \cdots\}$, we have $\mathbf{K} = \mathbf{AN}$, where $\mathbf{A}_{ji} = c_{j,i}$ is the matrix for coefficients. For simplicity, we consider training data from each domain will be *distributed* to the skills such that $\forall i, \sum_j N_i = 1$. This gives the amount of total training data from all domains is identical to the amount of total equivalent data for all skills, $\sum_j K_j = \sum_i N_i$. For a training data scale $N = \sum_i N_i = \sum_j K_j$, define optimal skill data composition $\mathbf{K}^* = diag\{K_a^*, K_b^*, \cdots\}$ as the minimizer of $\mathcal{L}$, given as $\mathbf{K}^* = \arg\min_{\sum_j K_j = N} \ell_0 + \beta_a \cdot K_a^{-\gamma_a} + \beta_b \cdot K_b^{-\gamma_b} + \cdots$. Theoretically, there can be an infinite number of latent skills. For analysis, we consider a finite number of $k$ independent skills *most important* for the evaluation. This can considered as performing Principal Components Analysis (PCA) with orthogonal transformation and selecting the first $k$ independent components. We consider the standard scenario with an equal number of relevant skills and data domains where $k = m$ and $\mathbf{A}$ is a square matrix with full rank. This describes the case where this optimization problem is well-defined. We discuss in App. B.2 what will happen in other scenarios. In this case, $\mathbf{A}$ is invertible and the corresponding optimal training data composition for $\mathbf{K}^*$ can be given as $\mathbf{N}^* = \mathbf{A}^{-1}\mathbf{K}^*$.

We provide the following theorem, which states that for the scenario described above, optimal training data composition scales in exponential-style functions with training data quantity and can be directly predictable from that of smaller scales *without needing to identify the latent skills*.

**Theorem 2** (Scaling Latent Skills). *Consider the evaluation is composed of a number of independent sub-tasks ("latent skills") where each of them observes a power law scaling law relationship with the amount of data contributing to this task ("equivalent data size"). Namely,*

$$\mathcal{L} = \ell_0 + \beta_a \cdot K_a^{-\gamma_a} + \beta_b \cdot K_b^{-\gamma_b} + \beta_c \cdot K_c^{-\gamma_c} + \cdots$$

*where scalar $K_j \geq 0$ denote equivalent data size for skill$_j$, and constants $(\beta_j, \gamma_j) \geq 0$ are coefficients associated with skill$_j$, respectively. Define diagonal matrices for training data composition $\mathbf{N} = diag\{N_1, N_2, \cdots\}$ and skill data composition $\mathbf{K} = diag\{K_a, K_b, \cdots\}$. Consider training data from each domain $D_i$ contributes to these skills to varying degrees, given as $\mathbf{K} = \mathbf{AN}$ where $\mathbf{A}_{ji} = c_{j,i}$ is the matrix for coefficients. Assume the amount of total training data from all domains is identical to the amount of total equivalent data for all skills, $\sum_j K_j = \sum_i N_i$. Assume there is a finite number of latent skills and data domains and $\mathbf{A}$ is a square matrix with full rank.*

*For a training data scale $N = \sum_i N_i = \sum_j K_j$, define optimal skill data composition $\mathbf{K}^* = diag\{K_a^*, K_b^*, \cdots\}$ as the minimizer of $\mathcal{L}$ s.t. $\sum_j K_j = N$ with corresponding optimal training data composition. If we have optimal data compositions $\mathbf{N}^*(N^{(1)}) = diag\{N_a^{(1)*}, N_b^{(1)*}, \cdots\}$ where its corresponding skill data composition $\mathbf{K}^{(1)*} = diag\{K_a^{(1)*}, K_b^{(1)*}, \cdots\} = \mathbf{AN}^*(N^{(1)})$ minimizes $\mathcal{L}$ s.t. $\sum_j K_j = \sum_i N^{(1)*} = N^{(1)}$, and $\mathbf{N}^*(N^{(2)}) = diag\{N_a^{(2)*}, N_b^{(2)*}, ...\}$ where its corresponding skill data composition $\mathbf{K}^{(2)*} = diag\{K_a^{(2)*}, K_b^{(2)*}, ...\} = \mathbf{AN}^*(N^{(2)})$ minimizes $\mathcal{L}$ s.t. $\sum_j K_j^{(2)*} = \sum_i N^{(2)*} = N^{(2)}$ where $N^{(2)} \neq N^{(1)}$, then, other optimal data compositions $\mathbf{N}^*(N^{(3)}) = diag\{N_a^{(3)*}, N_b^{(3)*}, ...\}$ where the corresponding skill data composition $\mathbf{K}^{(3)*} = diag\{K_a^{(3)*}, K_b^{(3)*}, \cdots\} = \mathbf{AN}^*(N^{(3)})$ minimizes $\mathcal{L}$ s.t. $\sum_j K_j^{(3)*} = \sum_i N^{(3)*} = N^{(3)}$ where $N^{(3)} \neq N^{(2)} \neq N^{(1)}$ must satisfy*

$$\mathbf{N}^*(N^{(3)}) = \mathbf{N}^*(N^{(2)})[(\mathbf{N}^*(N^{(1)}))^{-1}\mathbf{N}^*(N^{(2)})]^k$$

*for some $k \in \mathbb{R}^+$.*

*Proof.* By definition, we have

$$\mathbf{AN}^*(N^{(1)}) = \mathbf{K}^{(1)*}, \quad \mathbf{AN}^*(N^{(2)}) = \mathbf{K}^{(2)*}, \quad \mathbf{AN}^*(N^{(3)}) = \mathbf{K}^{(3)*}$$

From results of Theorem 1 in Section 3.2.2, we have

$$\mathbf{K}^{(3)*} = \mathbf{K}^{(2)*}[(\mathbf{K}^{(1)*})^{-1}\mathbf{K}^{(2)*}]^k$$

for some $k \in \mathbb{R}^+$, which gives

$$\mathbf{AN}^*(N^{(3)}) = (\mathbf{AN}^*(N^{(2)}))[(\mathbf{AN}^*(N^{(1)}))^{-1}\mathbf{AN}^*(N^{(2)})]^k$$

Since $\mathbf{A}$ is invertible and $\mathbf{N}$ and $\mathbf{K}$ are diagonal matrices, naturally,

$$(\mathbf{AN}^*(N^{(1)}))^{-1} = (\mathbf{N}^*(N^{(1)}))^{-1}\mathbf{A}^{-1}$$

and we have

$$\mathbf{AN}^*(N^{(3)}) = \mathbf{AN}^*(N^{(2)})[(\mathbf{N}^*(N^{(1)}))^{-1}\mathbf{A}^{-1}\mathbf{AN}^*(N^{(2)})]^k = \mathbf{AN}^*(N^{(2)})[(\mathbf{N}^*(N^{(1)}))^{-1}\mathbf{N}^*(N^{(2)})]^k$$

This directly gives

$$\mathbf{N}^*(N^{(3)}) = \mathbf{A}^{-1}\mathbf{AN}^*(N^{(2)})[(\mathbf{N}^*(N^{(1)}))^{-1}\mathbf{N}^*(N^{(2)})]^k = \mathbf{N}^*(N^{(2)})[(\mathbf{N}^*(N^{(1)}))^{-1}\mathbf{N}^*(N^{(2)})]^k$$

which completes the proof.

The above result does not require identifying the latent skills or observing skill data compositions $\mathbf{K}$. Rather, the theorem gives that as long as the coefficient matrix $\mathbf{A}$ is invertible, the scaling of $\mathbf{N}$ complies to the same scaling law as in Sec. 3.2.2. $\square$

*Remark* 2 (what happens when $\mathbf{A}$ is not invertible.). In general, if $\mathbf{A}$ is not invertible, scaling for optimal training data composition is not directly predictable. Specifically, if $\mathbf{A}$ does not have full rank, there exists redundant domains/data sources where their contribution to the skills are identical/exact multipliers of each other. Some data sources may not be needed at any scale; if $\mathbf{A}$ has more rows than columns (more domains than skills), this suggests multiple training data compositions can achieve the same skills data composition and the optimal training data compositions are non-unique (infinitely many). If $\mathbf{A}$ has more columns than rows (more skills than domains), this means there are too many skills to optimize for. No optimal training data composition exists and one has to make trade-offs. If this is relevant to the practical needs, training data may be processed with additional techniques such as clustering and split into more different domains.

## C    Experimental Details and Additional Results for Section 4, Evaluation

### C.1    Experimental Details on GPT-2 Large (774M)

**Evaluation** We test the perplexity on the held-out dataset, comprising 10K samples each from the 7 domains. For downstream tasks, we include: BoolQ (Clark et al., 2019) (zero-shot), HellaSwag (Zellers et al., 2019) (zero-shot, 10-shot), PIQA (Bisk et al., 2020) (zero-shot), TruthfulQA (Lin et al., 2021) (zero-shot), PubMedQA (Jin et al., 2019) (10-shot), CrowsPairs (Nangia et al., 2020) (25-shot), and ARC-Easy (Clark et al., 2018) (zero-shot). Additionally, BBH Novel Concepts (Srivastava et al., 2022) task is added to the aggregated results for models trained beyond 10B tokens, making a total of 9 tasks. We select tasks that ensure the model's performance surpasses random guessing, spanning from question answering and commonsense inference to bias identification and scientific problem solving. These tasks provide a comprehensive assessment of model performance (Mehta et al., 2024; Gadre et al., 2024). We adopt the evaluation framework from (Gao et al., 2021).

**Baselines** We report results for our methods (DDO and AutoScale ) and 6 baselines–UNIFORM, LLAMA WEIGHTS (curated), DOREMI (LLaMA weights initialization), DATA MIXING LAWS FROM (YE ET AL., 2024), DOREMI from Xie et al. (2024) (uniform initialization), and REGMIX from Liu et al. (2024). Uniform weights uniformly sample data from all domains, resulting in the same number of training tokens from each domain. LLaMA weights are a set of curated domain weights heuristically tuned for training LLaMA-1/2 models. We implemented DOREMI proposed in (Xie et al., 2024). DOREMI trains two smaller-scale auxiliary models (proxy models). First, a reference model is trained with the dataset's original domain weights, which are the LLaMA weights for RedPajama dataset. Then, optimized domain weights are obtained by using a proxy model to minimize the worst-case excess loss across different domains. We train both auxiliary models for 50K steps. Implementation details are available in App. C.3. Besides, we compare with 2 domain weights from existing literature, which are optimized on the same dataset, RedPajama, with similar Decoder-only LMs. DATA MIXING LAWS (Ye et al., 2024) first performs a grid search on the space of possible data mixtures and records evaluation loss for proxy models trained on these mixtures. Then, the loss is interpolated with exponential functions to find the optimal domain weights for the proxy model. DOGE (Fan et al., 2023) also implements DOREMI (Xie et al., 2024) with auxiliary models trained for 50K steps but with the reference model trained with uniform weights. REGMIX (Liu et al., 2024) first trains an array of smaller, proxy models on different data mix and small data scales, abd fits a regression model between domain weights and evaluation loss. Then, the fitted regression model is used to predict the evaluation loss for all feasible domain weights to find the best-performing weights. We use the same pairs of domain weights and evaluation loss DDO used in optimizing domain weights for 774M Decoder-only LMs at 0.3B tokens to fit REGMIX's LightGBM regressor. The fitted LightGBM model is then used to optimize the evaluation loss over domain weights. We evaluate the model trained on these domain weights to present a complete landscape.

**Model Training** GPT-2 Large is a variant of the GPT-2 architecture, featuring an embedding dimension of 1280, 36 transformer layers, and 20 attention heads. We rely on the Hugging Face Transformers library for implementation (Wolf et al., 2019). Specific training hyperparameters are detailed in Table 6.

**Dataset Details** The RedPajama dataset is available at: https://huggingface.co/datasets/togethercomputer/RedPajama-Data-1T. The 7 domains involved are characterized as follows:

- **Commoncrawl**: A vast repository of web-crawled data, providing a heterogeneous mix of internet text.

- **C4**: The Colossal Clean Crawled Corpus, filtered to remove low-quality content, thus ensuring the reliability and cleanliness of the data.

| Architecture | gpt2 |
|---|---|
| Optimizer | AdamW |
| Tokenizer Vocabulary Size | 50257 |
| Batch Size Per Device | 1 |
| Gradient Accumulation Steps | 10 |
| Maximum Learning Rate | 2e-4 |
| LR Schedule | Linear |
| Weight Decay | 1e-2 |
| Warm-up Ratio | 10% |
| Epochs | 3 |
| GPU Hardware | 8x NVIDIA A100/8x NVIDIA H100 |

Table 6: The list of hyperparameters for GPT-2 Large pretraining.

- **GitHub**: This domain includes a compilation of publicly available code repositories, offering a rich source of syntactic and semantic patterns inherent in programming languages.

- **Books**: A collection of textual content from published books, providing diverse narrative styles and complex character developments.

- **ArXiv**: Comprising scientific papers primarily from the fields of physics, mathematics, computer science, and quantitative biology, this domain offers high-quality, scholarly content.

- **Wikipedia**: A well-organized and meticulously curated dataset of encyclopedia articles, delivering a broad spectrum of knowledge across multiple disciplines. We only use English samples with 'en' in meta-data.

- **StackExchange**: This domain captures a variety of user-generated content from discussions and question-answer sessions across numerous technical topics.

Given copyright restrictions with the Books domain on Hugging Face, we have opted for an alternative source available at https://yknzhu.wixsite.com/mbweb.

For each domain, we ensure only samples with more than 1000 characters are retained. For each sample, the first 1000 characters are truncated, with the exception of the ArXiv and GitHub domains where we randomly extract a continuous block of 1000 characters. For the Wikipedia domain, we keep only those samples that are in English. Samples are selected without replacement, based on the computed data volume for each domain. Additionally, for each domain, a held-out dataset comprising 10K samples is reserved to evaluate the perplexity of the pretrained model.

## C.2 Experimental Details on BERT (110M)

We evaluate the model's MLM loss on held-out validation datasets, comprising 10K samples each from the 5 training domains. Additionally, as an auxiliary evaluation, we test the MLM loss on 3 non-training held-out domains. To be consistent with the perplexity loss used in CLM, we report the exponential cross-entropy loss for MLM. We evaluate the model's task performance on GLUE benchmark (Wang et al., 2018) (with 8 diverse tasks for natural language understanding (NLU)) and SQuAD (Rajpurkar et al., 2016) (a large-scale QA dataset). Uniform weights are used as the baseline.

**Model Training** We employ the BERT-base-uncased model from the Hugging Face Transformers library. Originally, BERT's pretraining scheme involved MLM and next sentence prediction (NSP); however, in our experiments, we exclusively utilize MLM. Detailed training hyperparameters can be found in Table 7.

**Dataset Details** The 5 domains of training data utilized are listed as follows:

| Architecture | bert-base-uncased |
|---|---|
| Max Token Length | 300 |
| Mask Token Percentage | 15% |
| Optimizer | AdamW |
| Batch Size Per Device | 12 |
| Devices | 4 |
| Maximum Learning Rate | 1e-4 |
| LR Schedule | Linear |
| Weight Decay | 1e-2 |
| Warm-up Steps | 3000 |
| Epochs | $1 \sim 4$ |
| GPU Hardware | 4x NVIDIA RTX A6000 |

Table 7: The list of hyperparameters for BERT pretraining.

- **Amazon Reviews**: A compilation of customer reviews from Amazon, widely utilized in sentiment analysis studies, available at: https://huggingface.co/datasets/amazon_us_reviews.

- **Arxiv**: Comprises 1.7 million articles from arXiv, available at: https://www.tensorflow.org/datasets/catalog/scientific_papers.

- **Books**: A corpus of 11,038 novels by unpublished authors across 16 genres, available at: https://yknzhu.wixsite.com/mbweb.

- **Wikipedia**: Offers datasets extracted from Wikipedia in various languages, available at: https://www.tensorflow.org/datasets/catalog/wikipedia. We only use English samples with 'en' in meta-data.

- **Open WebText Corpus (OWTC)**: A corpus of English web texts from Reddit posts, available at: https://skylion007.github.io/OpenWebTextCorpus/.

3 held-out non-training domains used in the evaluation include:

- **Pubmed**: Features 19,717 diabetes-related publications from the PubMed database, organized into three classes and linked by a network of 44,338 citations, available at: https://www.tensorflow.org/datasets/catalog/scientific_papers

- **News**: Comprises a significant collection of news articles derived from CommonCrawl, specifically from 5000 news domains indexed by Google News, available at: https://github.com/rowanz/grover/blob/master/realnews/README.md

- **GitHub**: A curated selection from the RedPajama dataset, this segment includes an array of open-source code projects, available at: https://huggingface.co/datasets/togethercomputer/RedPajama-Data-1T

### C.3 Implementation Details for Baselines

**Implementation details** We followed the official implementation[2] of DOREMI for our experiments. We evaluated two sets of reference domain weights: (1) the domain weights utilized in the LLaMA-2 paper Touvron et al. (2023) (referred to as LLaMA weights), and (2) uniform weights. Both the reference and proxy models have 120M parameters and are trained from scratch. We use GPT-2 tokenizer with a vocabulary size of roughly 50K. For LLaMA weights, we train each model for 20K, 50K and 200K steps for comparison. For uniform weights, we train each model for 10K, 20K and 50K steps. Refer to Table 8 for detailed hyperparameters. The effect of reference weights on the output DOREMI is discussed in Fig.9.

---

[2]https://github.com/sangmichaelxie/doremi

| | |
|---|---|
| Architecture | Decoder-only LM |
| Max Token Length | 1024 |
| Optimizer | AdamW |
| Batch Size Per Device | 8 |
| Devices | 8 |
| Maximum Learning Rate | 2e-4 |
| LR Schedule | Linear |
| Weight Decay | 1e-2 |
| Warm-up Steps | 3000 |
| Epochs | 1 |
| GPU Hardware | 8x NVIDIA RTX A6000 |

Table 8: The list of hyperparameters for DOREMI.

### C.4 Evaluation Details

**GPT/CLM** The following tasks are considered for downstream performance evaluation, in line with the setup from (Mehta et al., 2024; Gadre et al., 2024). For few-shot tasks, the demonstrations are sampled at random.

- **BoolQ** (Clark et al., 2019) consists of a question-answering format that requires binary yes/no answers.

- **HellaSwag** (Zellers et al., 2019) challenges models on their ability to make common-sense inferences.

- **PIQA** (Bisk et al., 2020) focuses on evaluating a model's commonsense reasoning regarding physical interactions.

- **TruthfulQA** (Lin et al., 2021) is designed to assess the ability of models to generate truthful and factual responses.

- **PubMedQA** (Jin et al., 2019) offers a dataset for evaluating question-answering in the biomedical domain.

- **CrowsPairs-English** (Nangia et al., 2020) tests models on their ability to identify and correct stereotypical biases in English text.

- **ARC-Easy** (Clark et al., 2018) presents a set of relatively simpler scientific reasoning questions, aimed at evaluating a model's basic understanding of scientific principles.

- **BigBench-Novel Concepts** (Srivastava et al., 2022) serves as a test of the model's creative abstraction skills, challenging it to make sense of scenarios that it could not have memorized during training.

**BERT/MLM** For each task, we conduct supervised fine-tuning on the corresponding training data and test the fine-tuned model on the validation data. The hyperparameters for supervised fine-tuning are given in Table 9.

| | |
|---|---|
| Architecture | bert-base-uncased |
| Max Token Length | 128 |
| Batch Size Per Device | 8 or 300 |
| Optimizer | AdamW |
| Devices | 4 |
| Maximum Learning Rate | 2e-5 or 5e-5 |
| Epochs | 3 |
| GPU Hardware | 4x NVIDIA RTX A6000 |

Table 9: The list of hyperparameters for supervised fine-tuning of BERT.

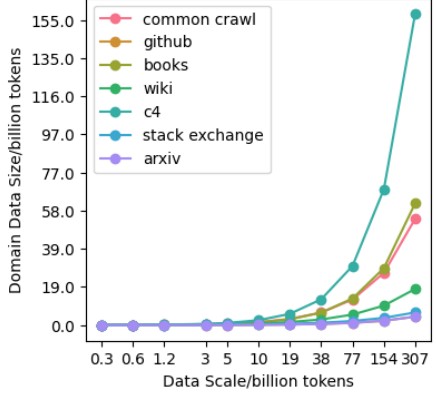
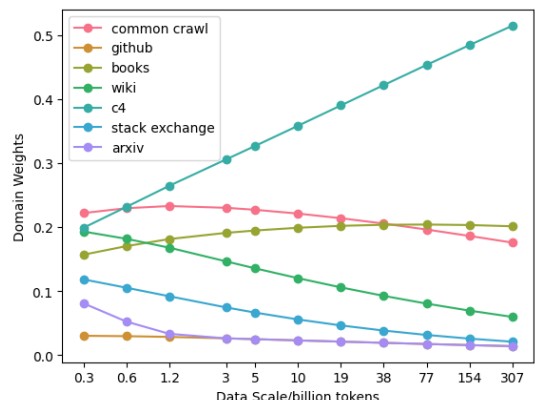

(a) AutoScale -predicted optimal data quantity for each domain as training data scales up.

(b) AutoScale -predicted optimal domain weights as training data scales up.

Figure 6: AutoScale -predicted domain weights for training 774M Decoder-only LMs. Optimal data quantity for each domain grows in exponential-style functions with training data scale (left) where data sources with diverse samples (e.g., `C4`) are upweighted relative to domains with standard format (e.g., `Wikipedia`).

### C.5 Additional Results on GPT-2 Large (774M)

Fig. 6 depicts AutoScale-predicted domain weights for training 774M Decoder-only LMs (GPT-2 Large). Optimal data quantity for each domain grows in exponential-style functions with training data scale (left) where data sources with diverse samples (e.g., `C4`) are upweighted relative to domains with standard format (e.g., `Wikipedia`).

Fig. 7 shows that when training on up to 5B tokens, AutoScale -predicted weights decreases val loss at least 25% faster than any baseline with up to 37% speed up.

Fig. 8 visualizes domain weights used for training `GPT-2 Large`, given by different methods.

Fig. 9 visualizes DOREMI optimized domain weights with different reference weights and training steps. Training proxy/reference models for different steps gives different weights. It is unclear which weights are optimal. DOREMI recommends 200k steps, which equals >100B tokens in the default setup. Since optimization was conducted relative to the reference weights, reference weights have a profound impact on DOREMI's output.

### C.6 Additional Results on BERT (110M)

Fig. 10(b) shows the results on fitting validation loss with power-law functions, directly approximating how loss changes with each domain's data quantity. Compared to `BERT` models trained with MLM (right), `GPT` models trained with CLM (left) demonstrate a much stronger response to domain reweighting. In final results, `GPT`/CLM achieved $> 2\times$ speedup margins relative to uniform weights compared to `BERT`/MLM.

Fig. 11 depicts the AutoScale -predicted domain weights for training `BERT`. It is evident that optimal data quantity for each domain grows in exponential-style functions with training data scale where data sources with diverse samples (e.g., `WebText`) are upweighted relative to domains with standard format (e.g., `ArXiv`).

Table 10 shows AutoScale notably improving training efficiency for `BERT` models on all scales–even for a considerably large scale, 288k steps, the speedup margin remains visible.

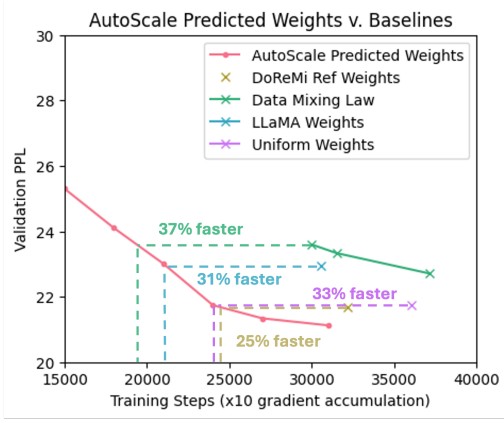 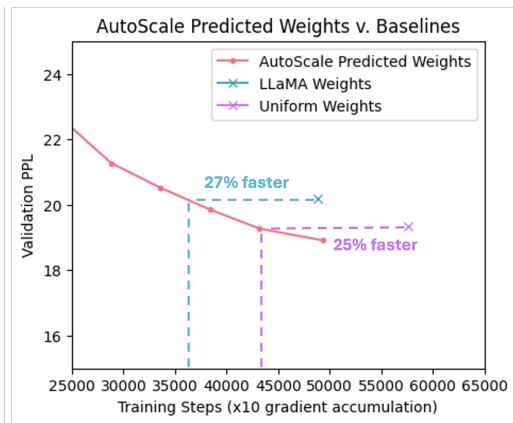

(a) Training Decoder-only LMs for 3B tokens.   (b) Training Decoder-only LMs for 5B tokens.

Figure 7: AutoScale -predicted weights decreases val loss at least 25% faster than any baseline with up to 37% speed up. Despite LLaMa weights being very different from uniform weights, they yield highly similar training efficiency at these data scales.

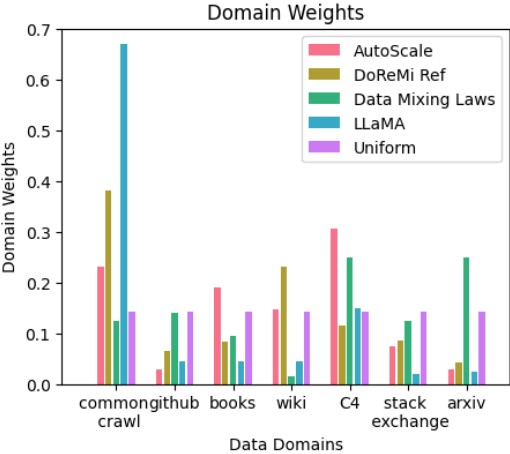

Figure 8: Domain Weights used for training 774M Decoder-only LMs for 3B tokens. (Domain weights for DATA MIXING LAWS and DOREMI are from references (Ye et al., 2024) and (Fan et al., 2023), respectively, which are implemented on the same datasets/data domains with highly similar model architecture/model size/tokenizers.)

| Data Scale/steps | 18k | 36k | 72k | 144k | 288k |
|---|---|---|---|---|---|
| Final Loss (exp) | 38.32 | 16.94 | 10.97 | 8.13 | 6.30 |
| Steps Saved | 5k (28%) | 5k (14%) | 10k (14%) | 20k (14%) | 20k (10%) |

Table 10: AutoScale notably improving training efficiency for BERT models on all scales–even for a considerably large scale, 288k steps, the speedup margin remains visible.

## D   Extended Discussions

**Intuition for Stage 2 of** AutoScale**, Optimal mix projection.**   Consider a stylized case where each training domain is independent. In this scenario, the validation performance attributable to each domain scales with the amount of training data from that same domain. Domains exhibit **different scaling behaviors:**

- As the training budget increases, the loss associated with some domains (e.g., 'easier' domains like standard texts from Wikipedia) decreases rapidly at first and then

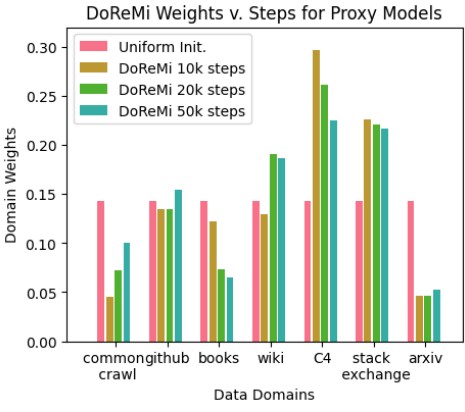

(a) with Uniform Reference Weights

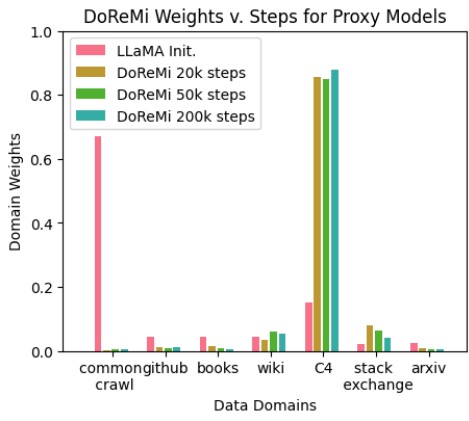

(b) with LLaMA Reference Weights (Default)

Figure 9: DOREMI with different reference weights and steps. Training proxy/reference models for different steps gives different weights. It is unclear which weights are optimal. DOREMI recommends 200k steps, which equals >100B tokens in the default setup. Since optimization was conducted relative to the reference weights, reference weights have a profound impact on DOREMI's output.

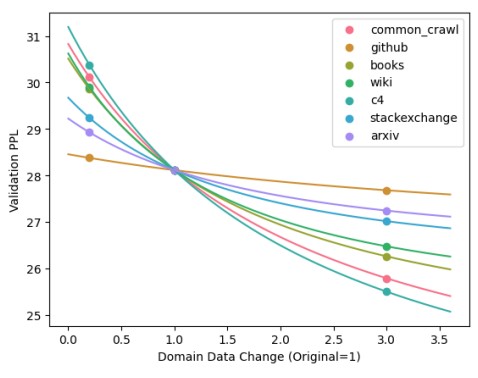

(a) 774M Decoder-only LMs (GPT-2 Large)

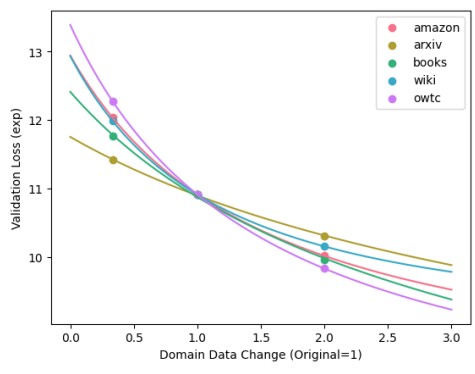

(b) Encoder-only LMs (BERT-case)

Figure 10: Fitting validation loss with power-law functions, directly approximating how loss changes with each domain's data quantity. Compared to BERT models trained with MLM (right), GPT models trained with CLM (left) demonstrate a much stronger response to domain reweighting. In final results, GPT/CLM achieved $> 2\times$ speed-up margins relative to uniform weights compared to BERT/MLM.

> plateaus. If the total training data budget is small, it is sensible to allocate more budget to such domains, as they offer a greater initial reduction in validation loss.
>
> - Conversely, for other domains (e.g., diverse sources like Common Crawl), the loss decreases more slowly but steadily with an increasing training budget, continuing to provide benefits even when the loss from other domains has plateaued.
>
> - With larger compute budgets, it becomes more beneficial to allocate resources to these steadily improving domains rather than adding more data to 'easy' domains that are already saturated. (As Figure 1 illustrates, optimal compositions are scale-dependent.) The key insight, which we formalize in a theorem, is that under certain independence assumptions, the evolution of optimal domain compositions with increasing training data budgets can be analytically derived from their individual scaling laws. This allows us to predict optimal compositions at larger data budgets.

We then relax the assumption of strict domain independence. We theoretically prove that if the validation data requires a number of independent 'latent skills,' and data from

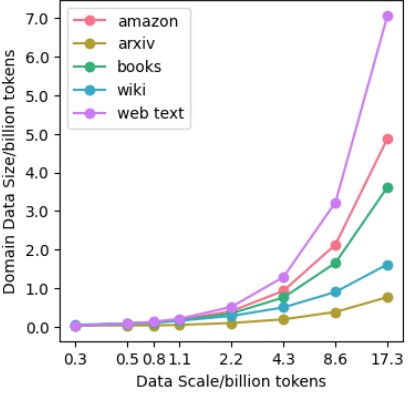

(a) AutoScale -predicted optimal data quantity for each domain as training data scales up.

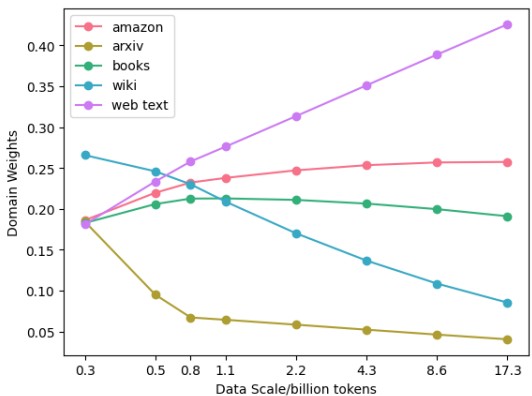

(b) AutoScale -predicted optimal domain weights as training data scales up.

Figure 11: AutoScale -predicted domain weights for training Encoder-only LMs (BERT). Optimal data quantity for each domain grows in exponential-style functions with training data scale (left) where data sources with diverse samples (e.g., WebText) are upweighted relative to domains with standard format (e.g., ArXiv).

each training domain contributes to one or more of these skills, our previous theory still holds in the same form. Thus, direct domain independence is not a strict requirement. This provides a complete theoretical grounding for our AutoScaletool, which captures the evolution of optimal domain weights with the training data budget and offers predictability for larger scales. Finally, our empirical evaluations demonstrate that the weights predicted by AutoScale offer clear advantages over domain weights optimized without considering this scale-dependence.the discussions and future work.

**Number of predictors (scaling law components)** We conducted ablation studies with DML on optimizing domain weights under a fixed compute dudget, following the same experiment procedure in Section 4.1 to pre-train GPT-2 Large models on the RedPajama dataset. According to DML's procedure, we fit a separate exponential function between domain weights and validation perplexity for each of the 7 domains, resulting in 7 separate predictors. When predicting for the average validation loss, we use each predictor to predict the validation perplexity for the respective domain and take the average of these predictions as the final output. When fitting on proxy training on random data mixtures, this prediction approach yields an average absolute relative error (AAR) = 4.46%. In comparison, similar to the treatment in the proposed DDO, we also fitted a single function between domain weights and the average validation perplexity using DML's exponential formula. We experimented with using this function to directly predict the average validation perplexity. When fitting on proxy training on random data mixtures, this prediction approach yields an AAR = 1.61%.

When using the fitted predictor to derive optimal data mixtures, we realized DML's formula does not allow optimize domain weights over a single predictor. For a validation domain, the predicted validation loss is given as the exponentiation of weighted average of domain weights plus some constant (Eq. (1) from Ye et al. (2024)). Since the domain weights are combined linearly before the exponentiation, if there is a single validation domain, minimizing the predicted loss will lead to a degenerate solution where all training data budgets is assigned to a single domain. Thus, it is necessary for DML to incorporate a number of separate predictors to introduce the crucial nonlinearity needed to model the interactions between different domains. The power-law formula in DDO is able to capture the nonlinearity in domain interactions in a single predictor, leading to lighter parametrization and enhanced efficiency

