# OpenReview forum: "AutoScale: Scale-Aware Data Mixing for Pre-Training LLMs"
_colmweb.org/COLM/2025/Conference — COLM 2025_

### Official Review · Reviewer_TYrQ · 2025-05-09

**Rating:** 6
**Confidence:** 3
**Ethics Flag:** 1

**Summary:**

This paper focuses on the scenario of data mixture for LLM pre-training. Inspired by scaling-law, it verifies that the optimal mixture rates are not uniform across different scales. By leveraging DDO and iteratively scaling based on the theoretical analysis, the paper proposes AutoScale, which seeks larger-scale data mixture at a lower cost using smaller models. Extensive experiments conducted on GPT-2 (774M) and BERT (110M) with ~10B tokens validate the effectiveness of the proposed method.

**Questions To Authors:**

1. Following the scaling-law, I am curious about whether the proposed method can be applied to model-size scaling as well?
2. Additional reference: BiMIX [1] also discusses data-mixing scaling in the context of pre-training, it is suggested to be included in the discussion and comparison in this paper.
3. Minor notation: In Theorem 1, the two variables \$N^\*\$ and N in the optimal domain allocation are somewhat difficult to distinguish. A different notation could enhance clarity.
4. Minor typo: In line 286, there should be a space between "AutoScale" and "consistently."
5. Minor figure refinement: In Figure 3, the image is not a vector graphic, the axis font is overly large, and the background color at '38%' obscures the subsequent line variation of AutoScale.

[1] A Bivariate Scaling Law for Language Model Pretraining

**Reasons To Accept:**

1. *Valuable Scenario:* This paper addresses an important, interesting, and practical problem of exploring large-scale data mixtures with minimal cost, which is crucial for LLM pre-training, given that trial-and-error can be an expensive process.
2. *Well-Grounded and Solid Clarification:* The insights presented in the paper are empirically driven, while the proposed method is theoretically grounded. The dual foundation enriches the content and provides solid support for the findings. Moreover, the conclusions align with existing discourse in the community, offering additional insights.
3. *Easy to Follow:* The paper features a clear structure and is well-written, making the methods easy to understand. The conclusions drawn are also clear, with comprehensive experiments that exhibit coherent logic.

**Reasons To Reject:**

1. *Stability and Robustness of the Method:* The stability of Stage 2 of the proposed method, as discussed in lines 219-225, under multi-round iterations has not been adequately considered. The statement regarding "… for any larger scale" and "… any desired accuracy" may be overly optimistic. While the initial scaling-up iterations might yield accurate results, the potential for error could gradually accumulate as the scale increases. This aspect has not been addressed in the paper.
2. *Experimental Scale and Model Selection:* Given that the data mixing scenario in the paper focuses on pre-training, the current volume of data and model-size is still insufficient to support prevailing standards in the field, despite the authors' explanations in lines 237-239. Although I understand that as a validation method, large-scale experiments may not be necessary, conducting experiments at a larger scale would enhance the reliability of the paper's conclusions (related to my previous concerns about stability).
   - The volume of data cited in the paper is limited to 10B tokens, which is quite small compared to the current mainstream pre-training requirements (for instance, the Llama3 series uses 15T tokens).
   - The models used in the study, GPT-2 (774M) and BERT (110M), are relatively outdated frameworks, and their sizes do not align with the scale of contemporary models.
3. *Iterative Scaling Curve:* Still considering Stage 2 of the proposed method, the iterative scaling approach allows for the evaluation of larger-scale performance. This raises the question of whether additional data points could be used to plot an exponential-like curve that describes the properties of mixture scaling. Utilizing this "ground truth" curve would facilitate a better assessment of the accuracy and effectiveness of this iterative generation process.

---

> ### Author Response · Authors · 2025-06-03
> **Authors' Response**
>
> Thanks for the effort reviewing this manuscript, and we appreciate your helpful feedback :) We provide the responses to the listed issues and questions below.
>
> ---
>
> ## Q1:  whether the proposed method can be applied to model-size scaling as well?
>
> **Re:** Technically, yes. Applying the data-scaling methodology to model-size scaling appears plausible. Scaling law papers [ref: https://arxiv.org/abs/2001.08361 , https://arxiv.org/abs/2203.15556 ] reported similar scaling trends between validation loss and training data budget/model size.
>
> This work has not explored this research problem. It is uncertain whether the combined effects of data and model-size scaling would be linear or if additional modeling would be required. We are happy to add to the discussions and future work.
>
> ---
>
> ## Q2: include BiMIX in the discussion and comparison in this paper
>
> **Re:** Thank you for suggesting this reference. BiMIX is indeed relevant to the scope of our work, and we are including it in the discussions on related work.
>
> There are a few major differences that distinguish our work from BiMIX:
>
> **1. Different Formulations:** BiMIX assumes independent scaling of each training domain, resembling an offline counterpart to ADO (https://arxiv.org/abs/2410.11820 ), which results in a model formulation quite different from ours. In contrast, our work (DDO) treats the validation dataset as an integral, holistic entity, rather than focusing on a sum of losses on individual training domains. As its name suggests, DDO (Direct Data Optimization) aims to "directly" minimize the loss on the entire validation dataset, irrespective of its specific content.
>
> **2. Focus and Evaluation:** BiMIX appears to focus on predictive ability rather than direct loss optimization, with its evaluation not directly reporting on overall loss reduction. The evaluation of BiMIX primarily includes exact match scores on three tasks, without reported validation loss or perplexity. **While BiMIX appears to perform comparably to RegMix, our experiments demonstrate that our proposed method, AutoScale, is clearly advantageous over RegMix in terms of accelerating the training process and improving validation loss.**
>
> ---
>
> ## Q3-Q5: notations, typos, and visualization.
>
> **Re:** We appreciate the helpful feedback. On notations, we are changing the data budget notation, currently denoted as N, to a different notation, tentatively $B$ (for budget), to avoid potential confusion with $N\^*$ to improve readability.
>
> We fixed the typo due to LaTeX formatting—thanks for the catch!
>
> Thanks for the comment on the visualization issue. We will replace the figures with vector graphic, adjust the data size and text positioning.
>
> ---
>
> We look forward to discussions with the reviewer. Please also let us know if there are any further questions or comments about this paper. We strive to consistently improve the paper and it would be our pleasure to have your precious feedback!

---

> > ### Comment · Reviewer_TYrQ · 2025-06-09
> >
> > Thank for addressing some of the concerns raised in my review. The authors' clarifications have partially alleviated my *Questions*, while I noted that the central *Weakness* remains unresolved.
> >
> > In general, this paper does present notable strengths, but the unaddressed concerns lead me to maintain my original recommendation rating (6).

---

> > > ### Author Response · Authors · 2025-06-10
> > > **Additional evidence to convince the reviewer in full support of publication**
> > >
> > > **Appreciating the reviewer for in support of publishing the paper, we would like to provide additional evidence to strengthen the contribution of this work and try to convince the reviewer in full support of publishing it :)**
> > >
> > > 1. **[On the Stability and Robustness of the Method]:** We think the reviewer's considerations are valid and we agree that the potential error could gradually accumulate as the scale increases. **On the positive side, we don’t think this will shadow the contribution of this work.**
> > >
> > >   - First, the scaling relationships predicted by AutoScale are derived from power-law relationships with the training data budget. A key property of our model is that the change in optimal domain weights **when scaling from, for example, 10B to 100B tokens is not proportionally more dramatic than the change observed when going from 1B to 10B tokens.** Indeed, our framework indicates that scaling up data budgets exponentially (e.g., to 100B or 1T tokens) primarily incurs **changes in domain weights that are closer to linear in scale, rather than the weights themselves changing exponentially.** We therefore anticipate that the relationships predicted by AutoScale will continue to provide valuable guidance even at these larger budgets.
> > >
> > >   - Further, **there has not been a way to accurately determine the optimal data composition for training LLMs at a large scale.** In current industrial practices (e.g., phi-4, llama-4, GPT-4), production-scale models are only trained once on the best estimate for the training data recipe, which is often determined by a coarse grid search among proxy training on a few candidate data mixtures. **The principled method developed in this work pioneers in the discovery of optimal training data mixture in a scale-aware manner, representing significant advancement over current practices.** Empirical results (with additional results and ablation studies provided in the comments under the summary review and in responses to Reviewer HTn5) clearly show AutoScale (this work) optimized weights to **deliver substantial performance gains over state-of-the-art approaches.**
> > >
> > > 2. **[On the Experimental Scale and Model Selection]:** We thank the reviewer for understanding the academic paper's role in developing validation methods. **Experiments conducted in this paper serve as prototypes to showcase the potential and impact the proposed approaches.** Despite the models size and data budget not comparable to scale of contemporary *commercial* models, the scale of experiments and investment in computation is *substantial* in terms of a research paper and academic contribution.
> > >
> > >   - Training each of these models for 10B tokens plus its evaluations took ~200 GPU hours on Nvidia A100 80GB GPUs, which costs ~$5k USD based on AWS pricing. With each training run repeated for at least 5-10 models for different baselines and ablation studies, **delivering main results in this manuscript cost ~$100k USD in compute based on AWS pricing** (not including the costs for development and debugging). **The magnitude of compute spending already exceeds the majority of academic efforts on this research problem and this work represents one of the best efforts in the community.**
> > >
> > > 3. **[On the Iterative Scaling Curve]**: We totally agree with the reviewer that having the "ground truth" curve would facilitate a better assessment of the accuracy and effectiveness of this iterative generation process. The prominent challenges are derived exactly from the two issues discussed above---**the lack of ground-truth solutions** and **the intractable computation costs at larger sizes**.
> > >
> > >   - Here, **we provide additional results of our best effort to bridge this gap.** At 0.3B, 0.6B, and 1.2B data budgets where we could afford repetitive model training, we optimize domain weights with the proposed DDO algorithm to determine "ground-truth" for optimal data mixtures. (With empirical studies conducted in the manuscript with additional results and ablation studies provided in the comments under the summary review and in responses to Reviewer HTn5, we show DDO to yield best-found solution among the state-of-the-art methods with the highest prediction precision. Thus, DDO-optimized weights serve as the **best-available proxy for the "ground-truth" of optimal domain weights.**) **Optimal domain data quantity for different training data scales shows remarkable high linearity $(R^2 = 0.998)$ on log-log plot, suggesting the shifting pattern can be well predicted by AutoScale's exponential-form iteration formula.**
> > >
> > > **These additional results and analysis serve as a further validation to the contribution of this work.** The combination of an advantageous design, a solid theoretical foundation, and substantial performance gains over state-of-the-art approaches ensures that **the publication of this work will meaningfully impact the advancement of the research field and its real-world applications.**

---

### Official Review · Reviewer_jtwN · 2025-05-11

**Rating:** 5
**Confidence:** 3
**Ethics Flag:** 1

**Summary:**

The authors develop and test a novel method, "AutoScale", for choosing an optimal mixture of domains to use in a training data set for pretraining an LLM at a given data scale, where data scale refers to the total number of tokens that will be present in the training data set.
The authors show that mixtures which are optimal at small scale often are not optimal at large scale, but testing multiple mixtures directly at large scale is prohibitively expensive. This is the problem that AutoScale is intended to address.

The AutoScale method rests on the assumption that the size of each domain, as a subset of a training data set, individually is governed by the empirical power scaling law identified by Kaplan et al. (2020), which defines a functional relationship between the amount of training data and validation loss for neural models. This assumption is then combined with the claimed original "Scale-Dependent Optimal Composition" theorem, which the authors prove in the paper (I do not have the expertise to verify the proof), to yield the "AutoScale" method for choosing the optimal mixture of domains in the training data set for a given training scale.

The authors present empirical results in which the application of the AutoScale method for choosing domain mixtures at large scale reduces training time and improves model performance compared to baselines.

**Questions To Authors:**

Do you anticipate that the scaling relationships predicted by AutoScale will continue to hold as larger budgets are used (e.g., greater than 100 B tokens)? Is it possible that a domain can appear to be headed toward saturation at small or medium scale but at large or very large scale the saturation does not continue or reverses?

How does AutoScale compare to recent methods that adjust the domain weight vector dynamically during training, instead of computing a fixed data mixture for a given training budget before starting a large-scale training run?
E.g., Jiang et al. (2024) [https://arxiv.org/abs/2410.11820], Luo et al. (2024) [https://arxiv.org/pdf/2411.14318]

**Reasons To Accept:**

This is a new method that can accelerate convergence and provide better scalability for training LLMs.

**Reasons To Reject:**

The paper is very dense, and much of the content required to understand the approach is in the appendix. I could not understand how the second stage of the method works or oven what the intuition behind it is. I understand that the authors are limited by the 9 page max, so maybe a conference paper is not the best venue for this. However, there are some redundancies between the intro and the lit review, so maybe the authors can tighten these sections and use the freed up space to explain at least the intuition behind stage 2.

---

> ### Author Response · Authors · 2025-06-03
> **Authors' Response (1/2)**
>
> Thanks for the effort reviewing this manuscript and we appreciate your genuine feedback :) We provide the responses to the listed issues and questions below.
>
> ---
>
> ## W1: dense elaborations
>
> **Re:** Thank you for the genuine feedback. **We strive to improve the readability and accessibility of the manuscript as a priority.**  Revisions are underway following the directions suggested by the review, **including condensing the introduction and literature review with additional discussions deferred to a new section on extended related works in the Appendix, strengthening the elaborations on insights and intuitions around methodology development, among a number of additional improvements** throughout the manuscript to provide better clarity. We believe the revisions further improve the quality of the manuscript and the contribution of this work.
>
> On a side note, as the reviewer pointed out, unlike purely empirical papers, this work presents **multiple original conceptual and technical contributions.** It challenges conventional beliefs and offers novel insights supported by validation results. The interconnected nature of these insights, methods, and results contributes to the manuscript's intricacy. We believe the technical depth of this manuscript is essential for readers and the community and our aim is to provide a reliable anchor for the field.
>
> ---
>
> ## W2: Intuition for Stage 2:
>
> **Re:**
> Consider a stylized case where each training domain is *independent*. In this scenario, the validation performance attributable to each domain scales with the amount of training data from that same domain. Domains exhibit **different scaling behaviors:**
>
> - As the training budget increases, the loss associated with some domains (e.g., 'easier' domains like standard texts from Wikipedia) decreases rapidly at first and then plateaus. If the total training data budget is small, it is sensible to allocate more budget to such domains, as they offer a greater initial reduction in validation loss.
> - Conversely, for other domains (e.g., diverse sources like Common Crawl), the loss decreases more slowly but steadily with an increasing training budget, continuing to provide benefits even when the loss from other domains has plateaued.
> - With larger compute budgets, it becomes more beneficial to allocate resources to these steadily improving domains rather than adding more data to 'easy' domains that are already saturated. (As Figure 1 illustrates, optimal compositions are indeed scale-dependent.)
>
> The key insight, which we formalize in a theorem, is that under certain independence assumptions, **the evolution of optimal domain compositions with increasing training data budgets can be analytically derived from their individual scaling laws.** This allows us to predict optimal compositions at larger data budgets.
>
> **We then relax the assumption of strict domain independence.** We theoretically prove that if the validation data requires a number of independent *'latent skills,'* and data from each training domain contributes to one or more of these skills, our previous theory still holds in the same form. Thus, direct domain independence is not a strict requirement. **This provides a complete theoretical grounding for our AutoScale tool, which captures the evolution of optimal domain weights with the training data budget and offers predictability for larger scales.**
>
> Finally, our empirical evaluations demonstrate that the weights predicted by AutoScale offer clear advantages over domain weights optimized without considering this scale-dependence.

---

> > ### Author Response · Authors · 2025-06-03
> > **Authors' Response (2/2)**
> >
> > ## Q: scaling to larger budgets (>100B), domain saturation or reverse at very large scales
> >
> > **Re:**
> > 1. The scaling relationships predicted by AutoScale are derived from power-law relationships with the training data budget. A key property of our model is that the change in optimal domain weights **when scaling from, for example, 10B to 100B tokens is not proportionally more dramatic than the change observed when going from 1B to 10B tokens.** Indeed, our framework indicates that scaling up data budgets exponentially (e.g., to 100B or 1T tokens) primarily incurs **changes in domain weights that are closer to linear in scale, rather than the weights themselves changing exponentially.** We therefore anticipate that the relationships predicted by AutoScale will continue to provide valuable guidance even at these larger budgets.
> >
> > 2. Regarding the possibility of saturation trends changing: A common understanding of foundation models (or LLMs) is that their substantial expressive capacity allows performance to improve according to established scaling laws (e.g.,[https://arxiv.org/abs/2001.08361 , https://arxiv.org/abs/2203.15556 ]), generally without saturating within currently explored budget ranges. **To our knowledge, existing research on optimizing domain weights has not explicitly modeled the eventual performance saturation of these models.** While model saturation at some future point is conceivable, **under the current trajectory of model scaling, it is plausible that data availability limits could become a more pressing constraint before intrinsic model saturation is reached** (e.g., Scaling Data-Constrained Language Models, https://arxiv.org/abs/2305.16264 ).
> >
> > While a detailed exploration of such effects at extreme scales is **beyond the primary scope of this work, this is indeed an interesting point.** We would be happy to incorporate discussions of these broader considerations in a revised manuscript.
> >
> > ---
> >
> > We look forward to discussions with the reviewer. It would be much appreciated if you could help review our responses and let us know if these address or partially address your concerns and if our explanations are heading in the right direction :)
> >
> > Please also let us know if there are any further questions or comments about this paper. We strive to consistently improve the paper and it would be our pleasure to have your precious feedback.

---

> > ### Comment · Reviewer_jtwN · 2025-06-09
> > **dense elaborations and stage 2**
> >
> > I think those changes are necessary, but since I cannot judge whether the authors will do them, I'm a bit hesitant to change my scores. (While I appreciate the explanation for stage 2, this really needs to be explained in the paper.)

---

> > > ### Author Response · Authors · 2025-06-10
> > > **Additional evidence to convince the reviewer in support of publication**
> > >
> > > We thank the reviewer for participating in the discussion and providing feedback on our responses. **While the reviewer appears to be on the edge of decisions, we would like to provide additional evidence to strengthen the contribution of this work and convince the reviewer in support of publishing it (even if not in full support :) ).**
> > >
> > > 1. **[Reviews will be public upon publication of this work. Hereby, the authors make a public pledge to diligently revise the manuscript as we outlined above.]** COLM does not allow updating the manuscript during rebuttal, which is the only way to provide evidence on improving the presentation. Typically, the exchange during the review/rebuttal process is requesting for clarification of factual problems or additional evidence. Besides, the listed presentation issues are not fundamental to the contribution of this work and **its revision is not a particular challenging task with unpredictable research outcomes.** The authors believe the delivery of planned revisions might not be the ground to the rejection vote. **We hope the reviewer will reconsider the overall rating of the manuscript considering the full value for its conceptual and empirical contributions and the potential scientific impact of its publication.**
> > >
> > > 2. **[During the rebuttal, we added results from substantial experiments and ablation studies on fine-grained comparisons with related work.]** Provided in the comments under the summary review and in responses to Reviewer HTn5, we show that comparing to concurrent works such as Data Mixing Laws (DML) and RegMix, our proposed DDO algorithm for optimizing domain weights under a fixed data budget (Stage 1) achieves the highest accuracy while requiring an order of magnitudes of proxy training. This showcases the advantageous efficiency and effectiveness of methods developed in this work. While predicting optimal domain weights for larger data budgets (Stage 2), AutoScale (this work) optimized weights led to superior results significantly outperforming all baselines. The theoretical analysis also visualizes key differences between methods developed in this work and concurrent efforts, pinpointing design issues for concurrent works such as Data Mixing Laws, RegMix and highlighting the essential advantages unique to our proposed methods.
> > >
> > > **These additional results and analysis serve as a further validation to the contribution of this work.** The combination of an advantageous design, a solid theoretical foundation, and substantial performance gains over state-of-the-art approaches ensures that **the publication of this work will meaningfully impact the advancement of the research field and its real-world applications.**

---

> ### Author Response · Authors · 2025-06-06
> **Rebuttal period ends soon–we anticipate your feedback!**
>
> Dear Reviewer jtwN,
>
> As the rebuttal/author discussion period ends in a few days, we sincerely look forward to your feedback. The authors are deeply appreciative of your valuable time and efforts spent reviewing this paper and helping us improve it.
>
> *In the responses, we provided thorough explanations to the listed issues, conducted additional experiments and ablation studies, and included in-depth discussions on comparisons with related works.* It would be very much appreciated if you could once again help review our responses and additional results and let us know if these address or partially address your concerns and if our explanations are heading in the right direction :)
>
> Please also let us know if there are any further questions or comments about this paper. We strive to consistently improve the paper and it would be our pleasure to have your precious feedback!
>
> Kind Regards,\
> Authors of Submission623

---

### Official Review · Reviewer_HTn5 · 2025-05-12

**Rating:** 5
**Confidence:** 5
**Ethics Flag:** 1

**Summary:**

Summary: many works on data mixing utilize proxy training runs (smaller model, fewer tokens) to determine the optimal data mixture at a larger scale. However, this paper shows that the optimal data mixture can change depending on the token budget. The paper proposes a two stage approach: first, a regression-based mixing algorithm that requires fewer proxy runs than Data Mixing Laws and RegMix; second, a theoretical analysis that leads to a way of scaling up the optimal mix at a small token budget to a larger token budget. Experiments show that AutoScale improves over RegMix and Uniform weights.

**Questions To Authors:**

See "Reasons to reject" above, in particular:

1) What are the downstream performance changes when using $w_1$ vs $w_2$?
2) How much does changing the random seed (in data) impact perplexity?
3) How much does a different fitting approach (using a different baseline mix and perturbation size) impact perplexity?
4) Can you report AAR on train/test mixes for DDO and RegMix? Can you report AAR when using a scaling law that has $N_0^{ij}$?
5) What happens when you use one stage of AutoScale and another stage of Data Mixing Laws? I am curious how Data Mixing Law's token budget scaling performs in conjunction with DDO.

**Reasons To Accept:**

- Data mixing is a critical step of the pretraining curation pipeline, and the current reliance on smaller and shorter proxy runs should be studied more closely, which this paper does.
- Generally thorough evaluation, including average test perplexity, downstream tasks, reporting the mixture weights, etc.
- Paper provides a rigorous theory for how to scale across token budgets and provides a motivating example for why this is important in Figure 1.

**Reasons To Reject:**

**Related work:** since this paper mentions how $\gamma_i$ changes the importance of each domain over time, online data mixing methods should also be discussed in the related work, perhaps as an alternative to determining how to scale mixes across token budgets. An incomplete list of these include: ODM (https://arxiv.org/abs/2312.02406), Skill-It (https://arxiv.org/abs/2307.14430), Aioli (https://arxiv.org/abs/2411.05735), and ADO (https://arxiv.org/abs/2410.11820).

**Effect of repeated documents:** for the paper's setting, scaling up the token budget to 10B should not result in any repeated documents for any mixture over RedPajama. However, in practice it is conventional to train for much longer and utilize all of the unique documents available at least once (i.e., the "data-constrained setting" of https://arxiv.org/abs/2501.11747). Repeating documents could significantly change the scaling law and theory of this work, although I do not think this significantly diminishes the paper's contributions.

**Statistical significance of Figure 1:** It is natural that N* @ 1.2B is going to be different from N* @ 0.3B, but it is unclear how much of this is due to noise. While Figure 4 shows that Figure 1's $w_1$ and $w_2$ are different---the weight for Github only slightly changes, while the weight for C4 changes significantly---it would be helpful to ground Figure 1's perplexity reductions by comparing it to other interventions, especially because the changes in perplexity reductions are not significant enough for DDO to perform worse than RegMix or uniform weights. Some suggestions:
- Compare the changes in downstream performance of $w_1$ vs $w_2$
- Measure how much changing the random seed (shuffling data/sampling "fresh" documents from each domain) impacts perplexity
- Measure how much a different fitting approach (i.e., using different baseline and perturbation) impacts perplexity

**Justification for scaling law:** in line 175, the approximation of $\mathcal{L}^v$ using a constant $N_0^i$ is interesting; why not make the $N_0^i$ term dependent on $w_1, w_2, \dots, w_{i-1}, w_{i+1}, \dots, w_m$? I.e., something of the form $\sum_{i=1}^m \big(\sum_{j \neq i}^m w_j \cdot N_0^{ij} + w_i \cdot N \big)^{-\gamma_i} + \ell_i$? Note that this reduces to your scaling law if $N_0^{ij} = N_0^i$ for all $j$, i.e., the equivalent data size is constant for each domain (I am skeptical that this is constant, since intuitively, e.g., CC and C4 interact with each other more than some other domain pairs)
- The result on AAR in line 275 should be explored more. Namely, reporting predicted loss on "training data" (baseline mix and perturbations) vs test data, and also reporting AAR for RegMix and Data Mixing Laws. A discrepancy in test data prediction accuracy between DDO and RegMix would justify why RegMix performs more poorly.

**More evaluation of Data Mixing Laws (Ye et. al.):** Data Mixing Laws has a two stage approach where it fits a mixing law and then uses a scaling law to determine the mix at a larger token budget. It would be interesting to try DDO + Data Mixing Law's token budget scaling law, and Data Mixing Law's mixing law + AutoScale's token budget scaling.

---

> ### Author Response · Authors · 2025-06-03
> **Authors' Response (1/2)**
>
> Thanks for the effort reviewing this manuscript and we appreciate your genuine feedback :) We enjoy the thoughtful exchange with the reviewer and respond to the listed issues and questions below.
>
> ---
>
> ## W1: online vs. offline data mixing methods
>
> **Re:**
> **1. [Offline and online methods represent two independent paths to approach data mixing problems in LM pre-training. Current practices are typically offline.]** Offline methods, such as DoReMi, Doge, RegMix, Data Mixing Laws (DML), and our proposed AutoScale, **decouple data curation from the final training process.** This separation allows for the distribution of model development workloads across different teams (e.g., data, model, and training) working in parallel. This approach is common in industrial settings for developing state-of-the-art LMs (e.g., Phi-4 [https://arxiv.org/abs/2412.08905 ], Llama4 [https://ai.meta.com/blog/llama-4-multimodal-intelligence/ ]). Notably, many offline methods like DoReMi (Google), DML (Shanghai AI Lab), and BiMIX (Alibaba) originate from industrial research, whereas current online methods (ODM, Skill-it, Aioli, ADO) are primarily academic contributions.
>
> **2. [Limitations of Online Methods in Modeling Training Dynamics.]** Offline methods treat the training process integrally, primarily adjusting domain weights and training budgets before training commences. In contrast, online methods (e.g., ODM, Skill-it, Aioli, ADO) adjust domain weights dynamically during the training process. ODM, for instance, models online domain weighting as a bandit problem and updates domain weights based on local gradients. Given that **the learning rate changes substantially throughout training**, the loss landscape can be considerably nonlinear and non-convex. **Consequently, a greedy approach offers no guarantee of global optimality. Skill-It primarily considers fine-tuning scenarios**, while Aioli extends this to small-scale LM pre-training. Aioli performs online estimation of the scaling law relationship between each pair of training and validation domains (in contrast to DML's offline estimation), while ADO only considers scaling laws for individual training domains. Beyond potential inaccuracies from online estimation and the omission of combinatorial domain effects, **the scaling law relationships these methods rely on typically hold for adjusting training data size within a fixed training pipeline** [ref: https://arxiv.org/abs/2001.08361 , https://arxiv.org/abs/2203.15556 ]. **They may not accurately track a training process, especially with a varying learning rate, undermining their theoretical grounding.**
>
> ---
>
> ## W2: Effect of repeated documents and “data-constrained setting”
>
> **Re:** This is an excellent point. We considered the ideas from [Scaling Data-Constrained Language Models, https://arxiv.org/abs/2305.16264 ] in full during the development of our research.
>
> The data constraint is indeed relevant for real-world LM pre-training, particularly for low-resource domains. Beyond constraints on domain weights, practical limitations include the amount of available data and the maximum number of repetitions tolerated before significant performance degradation. **To our knowledge, existing research on domain weighting has not explicitly modeled this factor.**
>
>
> **Despite this research gap,** [Scaling Data-Constrained Language Models, https://arxiv.org/abs/2305.16264 ] **offers positive evidence that current paradigms can still be effective under such constraints.** The paper indicates that repeating data for up to 3 epochs has almost no negative impact, and models can still train reasonably well with up to 7 epochs of repetition. This suggests that in practice, repeating data from some domains a few times can be acceptable without significant detriment to performance.
>
> ---
>
> ## W3 & Q1-Q4: statistical significance, additional studies and analysis
>
> **Re:** We appreciate these insightful questions regarding the statistical significance related to Figure 1 and the specific points Q1-Q4. *We are currently conducting additional experiments to address these questions directly. We anticipate that the results and a corresponding analysis will become available shortly.*

---

> > ### Author Response · Authors · 2025-06-03
> > **Authors' Response (2/2)**
> >
> > ## W4: Justification for scaling law, constant for "equivalent data size"
> >
> > **Re:** Thanks for the thoughtful comment. **This work considers the validation dataset as an integral part, as opposed to the sum of losses on each training domain. As the name, 'direct', suggests, DDO (Direct Data Optimization) aims to ``directly'' minimize the loss on the validation dataset regardless of its content.** Since there is only **one validation domain**, there only needs to be a single parameter for the equivalent data size between each training domain and the validation dataset. This framework is substantially different from existing works, including DML, Aioli, and ADO, and we found this treatment to be the key for both efficiency and effectiveness.
> >
> > - **In constrast, DML's formulation does not include the "equivalent data size" term to measure transfer effect from other domains [ref: Scaling laws for transfer, https://arxiv.org/abs/2102.01293 ].** In DML, there is only *one* constant term $c_i$ in the predictor for each validation domain $i$, which in fact represents the validation loss on this domain when the amount of training data from all domain is zero---in which case the loss should explode. **Fitting a constant term to an undefined variable appears questionable in design.**
> >
> > - Besides, DML/Aioli fits a separate predictor for each validation domain, **resulting in O(M×K) associated parameters**, where M and K represent training and validation domains (---in comparision,  DDO only fits a single predictor for the validation dataset with a linear number O(M) of parameters.) Fitting such a high number of parameters, especially with close or coupled domains where their influence is hard to distinguish, may be challenging and less unreliable. **This modeling approach serves as a good prototype combining the elements of scaling laws into the data mixing problem, but may fall short in both efficiency and accuracy.**
> >
> > - ADO only models scaling laws on each training domain and optimizes the aggregated loss on these training domains. **The scaling law for each training domain only considers itself and there is no explicit modeling for the transfer effect from other domains.** We share the sentiment that this aggressive approximation lacks sufficient grounding and appears inaccurate.

---

> > ### Author Response · Authors · 2025-06-04
> > **Additional Experiment Results and Analysis (RegMix)**
> >
> > ## W3 & Q1-Q4: statistical significance, additional studies and analysis
> >
> > Thanks for the interesting questions! We have conducted a range of additional experiments to ablate on the factors.
> >
> > 1. **[Downstream Performance] might not be reliable for proxy models trained with a small budget.** The models have not acquried capability over the threshold to produce meaningful performance. The limited signal may be drown in noise or biased by task selection. **Aligning with the line of research on scaling law and data mixing problems, we choose validation perplexity/loss as primary performance metric for fine-grained ablation studies.**
> >
> > 2. **[Statistical Significance]: We re-train the models at 0.3B and 1.2B data budgets with different random seeds and re-run the perplexity evaluations.** **Average validation perplexity remains stable**, scoring standard deviation=0.07, which is ~0.27% of average validation perplexity and <5% perplexity reduction (DDO-optimized weights compared to uniform weights).
> >
> >     - **Results in Figure 1 and subsequent observations remain statistically significant.** Following the helpful discussions, we are adding error bars for Figure 1 in the revised manuscript to the range of standard deviation and statistical significance of the results. Average absolute relative error (AAR) for DDO prediction on the new results is 0.32%. **Combined with results from previous training runs, the aggregated Average absolute relative error (AAR) for DDO prediction is revised from 1.00% to 0.66%.**
> >
> >
> > 3. **[AAR for RegMix]:** We train 8 models on random data mixtures drawn from a Dirichlet distribution. At 0.3B training data budget, the average validation perplexity from these proxy trainings ranges from 48.62 to 59.89. For comparison, at 0.3B training data budget, uniform weights and DDO-optimized weights led to average validation perplexity of 48.04 and 46.13, respectively.
> >
> >     - When fitting RegMix on validation perplexity from these 8 models, RegMix predictions record an Average absolute relative error (AAR) of 7.04%, failing to correctly capture the nonlinear effect for combining data from different domains. **Since the validation perplexity for all 8 proxy trainings is higher than that of uniform weights, in this case, RegMix or other regression-based methods cannot find data mixtures leading to lower validation perplexity.**
> >
> >     - We further added 15 more model trainings on data mixtures resembling a grid search on the simplex of domain weights. **When re-fitted on validation perplexity from these 23 models, RegMix prediction accuracy improves to an Average absolute relative error (AAR) of 3.16%, substantially improving compared to fitting on 8 model trainings.**
> >
> >     - **Based on the results and trend, we conjecture that with more proxy trainings covering more regions of the domain weights simplex, the prediction accuracy for RegMix will continue to improve**, helping to discover better data mixtures. At the extreme, a grid search could find global optimal solutions to any desired accuracy.  It is difficult to quantify how much prediction power is added by the regression model in RegMix (e.g., compared to interpolation with linear regression). *In comparison, DDO fitted on 15 proxy training already achieves an Average absolute relative error (AAR) of 0.66%.* ***The results clearly show the efficiency of the proposed DDO and its advantageous precision in discovering optimal data mixtures.***

---

> > > ### Author Response · Authors · 2025-06-05
> > > **Additional Results and Analysis: Data Mixing Laws (DML)**
> > >
> > > After carefully examining the formulations and successfully trouble-shooting certain numerical issues, we managed to fit DML on available proxy training datapoints.  **We conducted analysis on optimize domain weights under a fix data budget with DML and measured average absolute relative error (AAR) in its predictions.** We also studied DML’s behavior when **scaling up to larger training data budgets** and conducted **ablation studies on swapping DML’s components with our proposed DDO and AutoScale.** We present the results and analysis in the following comments.

---

> > ### Author Response · Authors · 2025-06-05
> > **DML Part 1: DML under a fixed data budget**
> >
> > ### a. Default DML formulation (separate predictors)
> >
> > According to DML’s procedure, **we fit a separate exponential function between domain weights and validation perplexity for each of the 7 domains, resulting in 7 separate predictors.** When predicting for the average validation loss, we use each predictor to predict the validation perplexity for the respective domain and take the average of these predictions as the final output. **When fitting on proxy training on 8 random data mixtures drawn from a Dirichlet distribution, this prediction approach yields an average absolute relative error (AAR) = 5.53%;** After further adding 15 more proxy training on 15 data mixtures resembling a grid search on the simplex of domain weights and **fitting DML on validation perplexity from these 23 models, this prediction approach yields an AAR = 4.46%.**
> >
> > ---
> >
> > ### b. Using a single predictor (as in DDO)
> >
> > **In comparison, similar to the treatment in the proposed DDO, we also fitted a single function between domain weights and the average validation perplexity using DML’s exponential formula.** We experimented with using this function to directly predict the average validation perplexity.
> > When fitting on proxy training on 8 random data mixtures drawn from a Dirichlet distribution, this prediction approach yields an **average absolute relative error (AAR) = 2.19%**; After further adding 15 more proxy training on 15 data mixtures resembling a grid search on the simplex of domain weights and **fitting DML on validation perplexity from these 23 models, this prediction approach yields an AAR = 1.61%.**
> >
> > ---
> >
> > ### c. Single-predictor approach is more accurate
> >
> > In general, when fitting on datapoints from the same number of proxy training, DML’s prediction accuracy under the default method (aggregating outputs from separate predictors) is on par with RegMix and lags behind the proposed DDO. Unlike RegMix, DML’s prediction accuracy does not appear to be heavily dependent on the number of datapoints used in the fitting. **Yet, when directly predicting average validation perplexity using a single predictor (the DDO formulation), DML’s prediction accuracy improves substantially, reducing prediction error by more than 50% and outperforming RegMix.** The improvements are especially favorable when fitting on fewer datapoints from proxy training, showing enhanced data efficiency.
> >
> > This aligns with our conjecture DML's heavily parameterized formulation might require additional effort to fit to reach a desirable accuracy. **DDO is developed with the insight that 'directly' optimizing over validation performance often leads to best outcomes.** This helps get rid of additional approximations which are potential sources of errors while easing the computation burden in fitting the predictor. The results above appear to support this observation.
> >
> > ---
> > ### d. DML's theoretical issues
> >
> > When using the fitted predictor to derive optimal data mixtures, we realized **DML's formula does not allow optimize domain weights over a single predictor.** For a validation domain $i$, the predicted validation loss $L_i$ is given as the exponentiation of weighted average of domain weights plus some constant $c_i$. Namely, Eq. (1) from DML:
> > $L_i=c_i+k_i\sum_j (t_{ij}r_j)$,
> > where $r_j$ is the proportion of data for training domain $j$ and $t_{ij}$ is its associated coefficient for validation domain $i$.
> > Since the domain weights are combined *linearly* before the exponentiation, if there is just a single validation domain, **minimizing the predicted loss will lead to a trivial solution where all training data budgets is assigned to a single domain**. Namely, $r_j = 1$ for the training domain with the largest coefficient $t_{ij}$ and $r_j = 0$ for all other domains. Thus, it is necessary for DML to incorporate a number of separate predictors to introduce the crucial nonlinearity needed to model the interactions between different domains.
> >
> > Grounded on empirical and theoretical insights from scaling laws research, **the power-law formula in DDO does not suffer from the same problem, capable of capturing the nonlinearity in domain interactions in a single predictor, leading to enhanced efficiency and effectiveness.** Besides, DML claims that
> > > "(the power law formula has) ill-posed properties that the function value blows up when the variable, mixture proportion in our case, approaches 0",
> >
> > which is not true. **The authors overlooked the effect of equivalent data size [https://arxiv.org/abs/2102.01293 ] from other domains.** Unless the authors consider training data from different domains
> > to have completely independent contributions to validation domains, when the amount of data from one training domain is zero, the data quantity in the power law function will be the equivalent data size measuring transfer effect of training data from other domains. The loss only "blows up" when the amount of training data from all domains is zero, in which case the loss is undefined.

---

> > ### Author Response · Authors · 2025-06-05
> > **DML Part 2: Optimizing data mixtures**
> >
> > We optimize domain weights for average validation perplexity. For DML, we fit separate predictors for each domain and solve for the domain weights leading to the lowest combined loss. The results are presented below,
> >
> > At 0.3B training tokens,
> > | Domain         | common_crawl | github | books | wiki  | c4    | stackexchange | arxiv |
> > |----------------|--------------|--------|-------|-------|-------|---------------|-------|
> > | DML (0.3B tokens):      | 0.50         | 0.02   | 0.03  | 0.10  | 0.30  | 0.02          | 0.02  |
> > | DDO (0.3B tokens):      | 0.22         | 0.03   | 0.17  | 0.20  | 0.21  | 0.10          | 0.07  |
> >
> > The optimized weights have similarities, both assigning lower weights to non-textual domains (github, arxiv). DML optimized weights assign most data budget to common_crawl and c4, while DDO optimized weights also assign budgets to books and wiki.
> >
> >
> > At 1.2B training tokens,
> > | Domain         | common_crawl | github | books | wiki  | c4    | stackexchange | arxiv |
> > |----------------|--------------|--------|-------|-------|-------|---------------|-------|
> > | DML (1.2B tokens):      | 0.21         | 0.02   | 0.06  | 0.04  | 0.60  | 0.03          | 0.03  |
> > | DDO (1.2B tokens):      | 0.23         | 0.03   | 0.19  | 0.17  | 0.26  | 0.09          | 0.03  |
> >
> > The trend remains, where DML optimized weights swap the budgets for common_crawl and c4 while further reducing budgets for other domains; DDO optimized weights change less dramatically and remain assigning budgets to books and wiki.

---

> > ### Author Response · Authors · 2025-06-05
> > **DML Part 3: Scaling up DML and comparison with AutoScale**
> >
> > We now predict for domain weights for 3B training budget.  For DML, we experimented with
> >
> > 1. using its default method. First fit data scaling laws to project results from proxy trainings at 0.3B and 1.2B data budgets to 3B, then fit its predictors on the projected datapoints to optimize domain weights; The results appear to be a continuation of the trend from 0.3B to 1.2B, where the optimized weights assigning more budget to C4 and less for others.
> >
> > | domains       |   DML (proj 3B) |   DDO+AutoScale (proj 3B) |   DML+AutoScale (proj 3B) |   DDO+SL(DML) (proj 3B) |
> > |:--------------|--------------:|-------------------:|------------:|---------------:|
> > | common_crawl  |         0.13  |               0.24 |        0.09 |           0.23 |
> > | github        |         0.03 |               0.03 |        0.02 |           0.03 |
> > | books         |         0.09  |               0.20 |        0.07 |           0.21 |
> > | wiki          |         0.04  |               0.16 |        0.02 |           0.11 |
> > | c4            |         0.64  |               0.30 |        0.74 |           0.32 |
> > | stackexchange |         0.03  |               0.08 |        0.03 |           0.08 |
> > | arxiv         |         0.04  |               0.02 |        0.03 |           0.03 |
> >
> > 2. AutoScale predicts weights at 3B from DDO-optimized weights at 0.3B and 1.2B. The optimized weights assign slightly more budget to C4 while reducing stackexchange.
> >
> > 3. Using AutoScale to predict weights at 3B from DML-optimized weights at 0.3B and 1.2B. **Due to the dramatic change from DML-optimized weights from 0.3B and 1.2B (e.g., the significant increase on C4), the predicted weights appear more extreme, assigning most weights to C4.**
> >
> > 4. Use scaling law to project proxy training datapoints at 0.3B and 1.2B to 3B (similar to DML’s data scaling), then use DDO to optimize domain weights at 3B. **The resulting weights are almost identical to those using DDO+AutoScale. The data scaling components in DML use the same formula (power laws) as AutoScale. It is not surprising the theorem developed for AutoScale still holds.**
> >
> > ---
> > ### Finally, we conduct actual model training on these weights for 3B tokens and evaluate the validation perplexity on each domain. The results are presented below.
> >
> > | domains       |   DML (proj 3B) |   DDO+AutoScale (proj 3B)  |   DDO+SL(DML) (proj 3B) |
> > |:--------------|--------------:|-------------------:|---------------:|
> > | common_crawl  |        28.904  |               25.598   |           **25.525**|
> > | github        |         8.303 |              **7.482**|           7.542 |
> > | books         |         34.003  |               29.162  |           **28.821** |
> > | wiki          |         23.835  |             **18.828** |           19.594 |
> > | c4            |         **30.808**  |              34.242  |           33.797 |
> > | stackexchange |         18.841  |            15.991 |                 **15.741** |
> > | arxiv         |         17.152  |              **16.558**  |                 16.629 |
> > | **Average**         |         23.121  |              21.123 |                 **21.093** |
> >
> > **Findings:**
> > 1. **DDO+AutoScale predicted weights** for 3B tokens (DDO+AutoScale (proj 3B), *methodology proposed in this work*) **significantly outperform DML-optimized weights** for 3B tokens (DML (proj 3B)), reducing average validation perplexity by 1.998 (or 9.47%). **This clearly demonstrate the performance edge of the proposed methods over DML.**
> >
> > 2. The weighted obtained by using scaling law to project proxy training datapoints at 0.3B and 1.2B to 3B (i.e., DML’s data scaling) and  optimizing domain weights with DDO at 3B (DDO+SL(DML) (proj 3B)) are almost identical to those obtained using DDO+AutoScale (DDO+AutoScale (proj 3B)). **And the validation performance for models trained on these weights are also near-identical**---the difference 0.03 or (0.142%) is well-within the margin of variance. **The resulting weights are almost identical to those using DDO+AutoScale. The data scaling components in DML use the same formula (power laws) as AutoScale. It is not surprising the theorem developed for AutoScale still holds.**
> > ---
> >
> > **This concludes the additional results and ablation studies.** In summary, DML’s exponential formula for optimizing domain weights appears less efficient or accurate, which is also inconsistent with its power law-based data scaling framework.
> > **Results from ablation studies further confirms the effectiveness for the power law scaling functions adopted in this work.** The proposed tools, DDO and AutoScale, with a consistent framework grounded in scaling laws research, delivering reliable results across a variety of setups and ablations.
> > ***Compared to existing research and concurrent works, this paper presents distinct contributions to the research field, benefiting Empirical use cases and theoretical analysis alike.***

---

> > ### Author Response · Authors · 2025-06-06
> > **Rebuttal period ends soon–we anticipate your feedback!**
> >
> > Dear Reviewer HTn5,
> >
> > As the rebuttal/author discussion period ends in a few days, we sincerely look forward to your feedback. The authors are deeply appreciative of your valuable time and efforts spent reviewing this paper and helping us improve it.
> >
> > *In the responses, we provided thorough explanations to the listed issues, conducted additional experiments and ablation studies, and included in-depth discussions on comparisons with related works.* It would be very much appreciated if you could once again help review our responses and additional results and let us know if these address or partially address your concerns and if our explanations are heading in the right direction :)
> >
> > Please also let us know if there are any further questions or comments about this paper. We strive to consistently improve the paper and it would be our pleasure to have your precious feedback!
> >
> > Kind Regards,\
> > Authors of Submission623

---

> > > ### Author Response · Authors · 2025-06-10
> > > **We addressed all of your concerns with comprehensive additional experiments and ablation studies :)**
> > >
> > > We thank the reviewer for the thoughtful reviews which have inspired the authors and motivated additional experiments and comprehensive ablation studies. **We provided careful responses to all of the listed issues and enjoyed the discussion with the reviewer. The analysis has led to additional findings and novel insights in addition to the original work.**
> > >
> > > In the additional results, we show that compared to concurrent works such as Data Mixing Laws (DML) and RegMix, our proposed DDO algorithm for optimizing domain weights under a fixed data budget (Stage 1) achieves the highest accuracy while requiring an order of magnitudes of proxy training. This showcases the advantageous efficiency and effectiveness of methods developed in this work. While predicting optimal domain weights for larger data budgets (Stage 2), AutoScale (this work) optimized weights led to superior results significantly outperforming all baselines. The theoretical analysis also visualizes key differences between methods developed in this work and concurrent efforts, pinpointing design issues for concurrent works such as Data Mixing Laws, RegMix and highlighting the essential advantages unique to our proposed methods.
> > >
> > > **These additional results and analysis serve as a further validation to the contribution of this work.** The combination of an advantageous design, a solid theoretical foundation, and substantial performance gains over state-of-the-art approaches ensures that **the publication of this work will meaningfully impact the advancement of the research field and its real-world applications.**

---

### Comment · Program_Chairs · 2025-04-03

This paper violates the page limit due to adding a limitation sections beyond the page limit. COLM does not have a special provision to allow for an additional page for the limitations section. However, due to this misunderstanding being widespread, the PCs decided to show leniency this year only. Reviewers and ACs are asked to ignore any limitation section content that is beyond the 9 page limit. Authors cannot refer reviewers to this content during the discussion period, and they are not to expect this content to be read.

---

### Author Response · Authors · 2025-06-02
**Review Summary, Comments, and Responses**

## Review Summary

**We are pleased that the paper has received thoughtful reviews, an overall positive reception, and unanimous recognition of its core contributions.** **Reviewers unanimously acknowledge that this work addresses a critical and valuable problem** in LLM pretraining by offering a novel approach with significant potential to accelerate convergence and improve training scalability (Reviewers HTn5, TYrQ, jtwN). **Reviews collectively confirm the paper's solid contribution, highlighting its well-grounded nature, thorough evaluation, rigorous theory, and clear exposition** (Reviewers HTn5, TYrQ). The theoretical grounding of AutoScale with the "Scale-Dependent Optimal Composition" theorem was consistently highlighted as a key strength providing a solid foundation for the method (Reviewers HTn5, TYrQ). The **thoroughness of the evaluation** demonstrating AutoScale's benefits over baselines was also well-received (Reviewers HTn5, TYrQ). Reviewer TYrQ also cited the paper's clarity and structure, which made the **core concepts accessible**.

---
## Responses to Comments

While acknowledging the conceptual novelty, technical contribution, solid grounding, and empirical effectiveness of our work, the reviews also included **valuable comments identifying areas for clarification and potential enhancement:**

* Reviewers expressed interest in **additional discussions comparing our work to other domain mixing methods** (e.g., offline methods like DML, RegMix, BiMIX, and **online methods** such as ODM, ADO, Aioli), covering conceptual differences and potential ablation studies.
  - In our responses, **we provide in-depth and comprehensive discussions** on these methods, demonstrating that AutoScale is substantially different from existing work and offers clear advantages in its conceptual design, theoretical grounding, and empirical results.

* Reviewers commented on the density of the manuscript and specifically suggested improvements to presentation and visualization.
  - We appreciate this genuine feedback and are committed to improving the manuscript's accessibility.

* Reviewers also suggested **discussions on broader scenarios**, such as data constraints, model scaling, and very large compute budgets.
  - We welcome these discussions on extended scenarios and elaborate on how our work has the potential to be applied to these broader cases, even if not its primary design focus.

We address these comments in detail under each individual review. *Some of our responses involve additional experiments; the corresponding results and analyses will be incorporated into the manuscript shortly as they become available.*

We thank the reviewers for their effort and constructive feedback. **We believe the resulting discussions have further improved the contribution of this work and the quality of the manuscript, allowing the paper to solidify its contributions to the field.**

---

> ### Author Response · Authors · 2025-06-04
> **Additional Experiment Results and Analysis (RegMix)**
>
> Following the suggestions from Reviewer HTn5, we have conducted a range of additional experiments to ablate on the factors.  Additional experiment results and analysis are provided below.
>
> 1. **[Statistical Significance]: We re-train the models at 0.3B and 1.2B data budgets with different random seeds and re-run the perplexity evaluations.** **Average validation perplexity remains stable**, scoring standard deviation=0.07, which is ~0.27% of average validation perplexity and <5% perplexity reduction (DDO-optimized weights compared to uniform weights).
>
>     - **Results in Figure 1 and subsequent observations remain statistically significant.** Following the helpful discussions, we are adding error bars for Figure 1 in the revised manuscript to the range of standard deviation and statistical significance of the results. Average absolute relative error (AAR) for DDO prediction on the new results is 0.32%. **Combined with results from previous training runs, the aggregated Average absolute relative error (AAR) for DDO prediction is revised from 1.00% to 0.66%.**
>
>
> 2. **[Average absolute relative error (AAR) for RegMix]:** We train 8 models on random data mixtures drawn from a Dirichlet distribution. At 0.3B training data budget, the average validation perplexity from these proxy trainings ranges from 48.62 to 59.89. For comparison, at 0.3B training data budget, uniform weights and DDO-optimized weights led to average validation perplexity of 48.04 and 46.13, respectively.
>
>     - When fitting RegMix on validation perplexity from these 8 models, RegMix predictions record an Average absolute relative error (AAR) of 7.04%, failing to correctly capture the nonlinear effect for combining data from different domains. **Since the validation perplexity for all 8 proxy trainings is higher than that of uniform weights, in this case, RegMix or other regression-based methods cannot find data mixtures leading to lower validation perplexity.**
>
>     - We further added 15 more model trainings on data mixtures resembling a grid search on the simplex of domain weights. **When re-fitted on validation perplexity from these 23 models, RegMix prediction accuracy improves to an Average absolute relative error (AAR) of 3.16%, substantially improving compared to fitting on 8 model trainings.**
>
>     - **Based on the results and trend, we conjecture that with more proxy trainings covering more regions of the domain weights simplex, the prediction accuracy for RegMix will continue to improve**, helping to discover better data mixtures. At the extreme, a grid search could find global optimal solutions to any desired accuracy.  It is difficult to quantify how much prediction power is added by the regression model in RegMix (e.g., compared to interpolation with linear regression). *In comparison, DDO fitted on 15 proxy training already achieves an Average absolute relative error (AAR) of 0.66%.* ***The results clearly show the efficiency of the proposed DDO and its advantageous precision in discovering optimal data mixtures.***

---

> > ### Author Response · Authors · 2025-06-05
> > **Additional Results and Analysis: Data Mixing Laws (DML)**
> >
> > **We conducted analysis on optimize domain weights under a fix data budget with DML and measured average absolute relative error (AAR) in its predictions.** We also studied DML’s behavior when **scaling up to larger training data budgets** and conducted **ablation studies on swapping DML’s components with our proposed DDO and AutoScale.** We present the selected results and analysis in the following comment with the full response under Reviewer HTn5.

---

> > > ### Author Response · Authors · 2025-06-05
> > > **DML under a fixed data budget: DDO's advantage by design**
> > >
> > > ### a. Default DML formulation (separate predictors)
> > >
> > > According to DML’s procedure, **we fit a separate exponential function between domain weights and validation perplexity for each of the 7 domains, resulting in 7 separate predictors.** When predicting for the average validation loss, we use each predictor to predict the validation perplexity for the respective domain and take the average of these predictions as the final output. **When fitting on proxy training on 8 random data mixtures drawn from a Dirichlet distribution, this prediction approach yields an average absolute relative error (AAR) = 5.53%;** After further adding 15 more proxy training on 15 data mixtures resembling a grid search on the simplex of domain weights and **fitting DML on validation perplexity from these 23 models, this prediction approach yields an AAR = 4.46%.**
> > >
> > > ---
> > >
> > > ### b. Using a single predictor (as in DDO)
> > >
> > > **In comparison, similar to the treatment in the proposed DDO, we also fitted a single function between domain weights and the average validation perplexity using DML’s exponential formula.** We experimented with using this function to directly predict the average validation perplexity.
> > > When fitting on proxy training on 8 random data mixtures drawn from a Dirichlet distribution, this prediction approach yields an **average absolute relative error (AAR) = 2.19%**; After further adding 15 more proxy training on 15 data mixtures resembling a grid search on the simplex of domain weights and **fitting DML on validation perplexity from these 23 models, this prediction approach yields an AAR = 1.61%.**
> > >
> > > ---
> > >
> > > ### c. Single-predictor approach is more accurate
> > >
> > > In general, when fitting on datapoints from the same number of proxy training, DML’s prediction accuracy under the default method (aggregating outputs from separate predictors) is on par with RegMix and lags behind the proposed DDO. Unlike RegMix, DML’s prediction accuracy does not appear to be heavily dependent on the number of datapoints used in the fitting. **Yet, when directly predicting average validation perplexity using a single predictor (the DDO formulation), DML’s prediction accuracy improves substantially, reducing prediction error by more than 50% and outperforming RegMix.** The improvements are especially favorable when fitting on fewer datapoints from proxy training, showing enhanced data efficiency.
> > >
> > > This aligns with our conjecture DML's heavily parameterized formulation might require additional effort to fit to reach a desirable accuracy. **DDO is developed with the insight that 'directly' optimizing over validation performance often leads to best outcomes.** This helps get rid of additional approximations which are potential sources of errors while easing the computation burden in fitting the predictor. The results above appear to support this observation.
> > >
> > > ---
> > > ### d. DML's theoretical issues
> > >
> > > When using the fitted predictor to derive optimal data mixtures, we realized **DML's formula does not allow optimize domain weights over a single predictor.** For a validation domain $i$, the predicted validation loss $L_i$ is given as the exponentiation of weighted average of domain weights plus some constant $c_i$. Namely, Eq. (1) from DML:
> > > $L_i=c_i+k_i\sum_j (t_{ij}r_j)$,
> > > where $r_j$ is the proportion of data for training domain $j$ and $t_{ij}$ is its associated coefficient for validation domain $i$.
> > > Since the domain weights are combined *linearly* before the exponentiation, if there is just a single validation domain, **minimizing the predicted loss will lead to a trivial solution where all training data budgets is assigned to a single domain**. Namely, $r_j = 1$ for the training domain with the largest coefficient $t_{ij}$ and $r_j = 0$ for all other domains. Thus, it is necessary for DML to incorporate a number of separate predictors to introduce the crucial nonlinearity needed to model the interactions between different domains.
> > >
> > > Grounded on empirical and theoretical insights from scaling laws research, **the power-law formula in DDO does not suffer from the same problem, capable of capturing the nonlinearity in domain interactions in a single predictor, leading to enhanced efficiency and effectiveness.** Besides, DML claims that
> > > > "(the power law formula has) ill-posed properties that the function value blows up when the variable, mixture proportion in our case, approaches 0",
> > >
> > > which is not true. **The authors overlooked the effect of equivalent data size [https://arxiv.org/abs/2102.01293 ] from other domains.** Unless the authors consider training data from different domains
> > > to have completely independent contributions to validation domains, when the amount of data from one training domain is zero, the data quantity in the power law function will be the equivalent data size measuring transfer effect of training data from other domains. The loss only "blows up" when the amount of training data from all domains is zero, in which case the loss is undefined.

---

### Decision · Program_Chairs · 2025-07-08

**Decision:**

Accept

**Comment:**

The paper introduces AutoScale, a two-stage, scale-aware data composition framework for pre-training. Stage 1 fits a parametric model to predict the model's loss under different data compositions --- used to find an approximate best allocation with less budget. Stage 2 extrapolates the optimal composition to larger budgets without further retraining. The authors demonstrate that AutoScale accelerates convergence and improves downstream performance, achieving a 28% faster perplexity drop and up to ~38% speed gain when pre-training GPT-2 large.

The reviews were generally sensible and asked some insightful and valuable questions which generated productive discussions and valuable improvements to the paper. All discussions should be incorporated into the revisions of the submission. The paper includes some solid theoretical and empirical contributions which not all reviewers were comfortable to assess. I do not think the paper should be penalised because there is not a directly appropriate journal venue for this work. One reviewer did not respond to the discussion, but I would consider their queries extremely valuable and the author's response to be thorough and highly valuable for paper improvements. Reviewers highlight the significance, volume and importance of the contributions to the paper. My opinion is that this will be a valuable feature in a conference agenda.